# Severe outcomes of malaria in children under time-varying exposure

Pablo M. De Salazar [1,2] ✉, Alice Kamau[3], Aurelien Cavelan[1,2], Samuel Akech[3], Arthur Mpimbaza[4], Robert W. Snow [3,5,8] & Melissa A. Penny[1,2,6,7,8] ✉

In malaria epidemiology, interpolation frameworks based on available observations are critical for policy decisions and interpreting disease burden. Updating our understanding of the empirical evidence across different populations, settings, and timeframes is crucial to improving inference for supporting public health. Here, via individual-based modeling, we evaluate a large, multicountry, contemporary *Plasmodium falciparum* severe malaria dataset to better understand the relationship between prevalence and incidence of malaria pediatric hospitalizations - a proxy of malaria severe outcomes- in East-Africa. We find that life-long exposure dynamics, and subsequent protection patterns in children, substantially determine the likelihood of malaria hospitalizations relative to ongoing prevalence at the population level. Unsteady transmission patterns over a lifetime in children -increasing or decreasing- lead to an exponential relationship of hospitalization rates versus prevalence rather than the asymptotic pattern observed under steady transmission. Addressing this increase in the complexity of malaria epidemiology is crucial to update burden assessments via inference models that guide current and future policy decisions.

Assessing the burden of malaria life-threatening outcomes in populations at risk is a critically important step in evaluating and improving control efforts. Malaria mortality is challenging to measure accurately in the community[1] but remains a fundamental component of statistical-based interpolation from prevalence estimates, resulting in high uncertainty[2]. Inference of disease burden has been approached with different grades of sophistication, ranging from purely data-driven fits to multi-level mechanistic microsimulations[3–6]. Independent of the complexity of the approach, the ability of a model to generate accurate, robust, and valid malaria disease outcomes using exposure predictors, such as prevalence, requires (1) high-quality data as input from real-world observations, and (2) a comprehensive understanding and

identification of the key factors determining the relationship between exposure and clinical outcomes.

Severe, life-threatening malaria syndromes presenting to hospitals are a valuable proxy for malaria-related death among communities. High exposure rates in children at a very young age are known to offset the risk of severe clinical outcomes at older ages[7]. This leads to a characteristic asymptotic pattern between exposure and disease risk at the population level, consistent with consensual malaria theory and historical observations[4,7–9]. Recent work has assessed the empirical relationship between community prevalence and the risk of severe malaria syndromes, namely severe anemia, cerebral malaria, and respiratory distress among children in East Africa, based on the largest standardized *Plasmodium falciparum* malaria pediatric dataset

[1]Department of Epidemiology and Public Health, Swiss Tropical and Public Health Institute, Allschwil, Switzerland. [2]University of Basel, Basel, Switzerland. [3]Kenya Medical Research Institute (KEMRI)-Wellcome Trust Research Programme, Nairobi, Kenya. [4]Child Health and Development Centre, College of Health Sciences, Makerere University, Kampala, Uganda. [5]Centre for Tropical Medicine and Global Health, Nuffield Department of Clinical Medicine, University of Oxford, Oxford, UK. [6]Telethon Kids Institute, Nedlands, WA, Australia. [7]Centre for Child Health Research, University of Western Australia, Crawley, WA, Australia. [8]These authors contributed equally: Robert W. Snow, Melissa A. Penny. ✉e-mail: pablo.martinezdesalazar@swisstph.ch; melissa.penny@uwa.edu.au

available to date[10]. Findings show that the occurrence of these severe malaria outcomes in the population may relate differently to increasing community prevalence. Particularly, an asymptotic relationship was observed when predicting severe anemia over community prevalence, while an exponential relationship was favored when predicting a combined outcome comprising the three syndromes. Evaluating new sources of standardized data, such as this contemporary dataset, contextualized with historical sources of data[9,11] improves our understanding of the accuracy, robustness, and validity of the inference frameworks.

Here, we use a previously validated multi-level individual-based malaria model[12], OpenMalaria (https://github.com/SwissTPH/openmalaria/wiki), to systematically investigate clinical and epidemiological factors influencing the relationship between potentially life-threatening hospital malaria admissions among children upon a given observed community prevalence. We use community-based malaria hospitalization incidence rates in small catchment populations and adjusted for case under-ascertainment as an empirical proxy of the incidence of severe malaria outcomes. Our analysis framework interrogates standardized malaria data obtained in Sub-Saharan African time-sites within the 1990s throughout 2020[9–11,13], aiming to improve and update our understanding of the dynamics from *P. falciparum* malaria infection to the occurrence of severe outcomes in children.

## Results

Malaria admissions were assembled from individual records of 21 hospitals representing 35 time-sites in Kenya, Uganda, and Tanzania, among children resident in specific catchment areas -within a defined distance radius from the hospital where surveillance took place- excluding urban settings where it was possible to estimate single-year censused population estimates. We assume that the hospitalization rates per time-site represent a lower limit of the hospitalization incidence and can be reasonably comparable when further adjusting for case ascertainment. Cases were included if malaria was the primary cause of hospitalization, and those with underlying conditions were excluded[10,13]. We further used data on malaria hospitalization incidence obtained with similar approaches between 1992 and 1997 at seven hospital time-site locations in Kenya, The Gambia, and Malawi[9,11]. This allows us to compare and interpret our findings with a dataset that has been classically used for informing malaria inference models[4,8,14] aiming to estimate severe outcomes of malaria across populations.

For each of the time-sites in the above datasets, the average number of hospitalizations due to malaria among children three months to 9 years old per 1000 children per year were paired with age-diagnostic method standardized community prevalence estimates, as empirical *P. falciparum* Parasite Rates among children 2–10 years old ($ePf$PR$_{2-10}$). The data was obtained from community and school surveys undertaken during the period of hospital surveillance within the same catchment areas[9–11]. Further, we assessed the past exposure dynamics using catchment site-specific time-series of modeled age- and test-standardized parasite rates estimates, herein referred to as $mPf$PR$_{2-10}$. For each time-site of the contemporary dataset, annual $mPf$PR$_{2-10}$ estimates were obtained using a Bayesian hierarchical geospatial model detailed elsewhere[13,15]. In those time sites where modeled estimates were available for at least 7 past years ($n = 27$), and up to a maximum of 9 years, we computed the median $mPf$PR$_{2-10}$ of the time series. We assume that the median prevalence across the past 7–9 years roughly represents the cumulative past transmission to which the population of children up to 10 years of age have been long-life, which can then be compared to the empirical prevalence at the time of the survey to evaluate the gap between past and present-day transmission

Visual inspection of the empirical relationship between prevalence and hospitalization rates for the 35 time-sites included in the contemporary dataset does not suggest an asymptotic relationship of malaria hospitalization incidence across the $ePf$PR$_{2-10}$ range within the full dataset (Fig. 1a.). For illustration, we highlight four representative time-sites in Fig. 1a and alongside the $mPf$PR$_{2-10}$ time-series of these time-sites for the years prior to the collection of the empirical data (Fig. 1b). Three major patterns of time-varying transmission are depicted, showing (1) a substantial increase in the $mPf$PR$_{2-10}$(Apac A), (2) relative constant $mPf$PR$_{2-10}$ (e.g, Busia) (3) steady increase (Mubende B) and 4) substantial decrease in the $Pf$PR$_{2-10}$(e.g., Jinja B). As depicted in Fig. 1c, there is substantial change between the $ePf$PR$_{2-10}$ and the median value of the $mPf$PR$_{2-10}$ over the previous years for each of the time-sites comparing at least seven years and up to ten years of modeled past exposure estimates. The difference between the $ePf$PR$_{2-10}$ and the median value of past $mPf$PR$_{2-10}$ can be interpreted as the gap in past exposure relative to ongoing exposure. For those time-sites at the higher end of the current $ePf$PR$_{2-10}$ range (i.e., higher than the median empirical prevalence, 20%), exposure had primarily substantially increased or remained relatively stable (12 out of 14 sites). For those time sites at the lower end of the range (lower than 20%), exposure had decreased or remained relatively stable (13 out of 13 sites). All available $mPf$PR$_{2-10}$ time series for the 35-time sites are shown in Fig. S1[13,15].

Understanding the complex relationship between malaria exposure, immunity, and clinical outcomes across populations and time requires causal analytical frameworks that (a) can combine empirical observations with theory (b) can address multiple interacting causal effects, threshold dynamics, and interference (c) have generally accepted principles to build the models, populate and calibrate their parameters and test their predictions for avoiding misspecification. Individual-based models are amongst the few modeling tools that fulfill these requirements[1,2].

Open-Malaria (https://github.com/SwissTPH/openmalaria/wiki) is an multi-level individual-based model that includes several key features that allow to generate counterfactuals of the effect of malaria exposure on clinical disease under different scenarios of population structure, changing transmission, health-access, diagnostic thresholds, drug-efficacy, and other major malaria control and prevention interventions[3]. Random effects can be incorporated into the modeled processes and allow the inclusion of uncertainty and heterogeneity in the simulations. Relevant to our analyses, the framework encompasses submodels specifically parameterized to empirical data including (1) population structure[4,5]; (2) within-host dynamics of parasite burden and addressing the effect on single individuals of repeated infections in developing immunity and subsequent infections[6–8]; (3) disease progression[6,7] and health-seeking behavior including rates of individuals accessing health services, as well as time to diagnostic and treatment[4]; (4) efficacy of case-management including diagnostic sensitivity and specificity, first- and second-line treatment effectiveness and efficacy of hospitalization[4]; and (5) the effect of age-structured comorbidities on severe malaria outcomes upon infection[5]. Further details are provided in the Supplementary Note 2.

We iteratively interrogated the data under different sets of plausible parameterizations of our individual-based model, hereafter referred to as scenario analysis. The scenario analysis explores hypotheses of the impact of well-known determinants on disease risk and changes in contemporary disease risk compared to historical observations, including the deployment and availability of artemisinin-derivatives in primary- and hospital-care, improved treatment adherence, the reduction in the occurrence and progression of malaria-associated comorbidities. In addition to addressing the detailed major key changes between the historical and contemporary datasets, we further address the life-long exposure dynamics evidenced by the modeled community prevalence estimates (Fig. 1b and Fig. S1). The risk of malaria disease outcomes, including those severe, depends on immunity, which in turn depends on previous exposure dynamics. We hypothesize that the year-to-year variability of PfPR could strongly

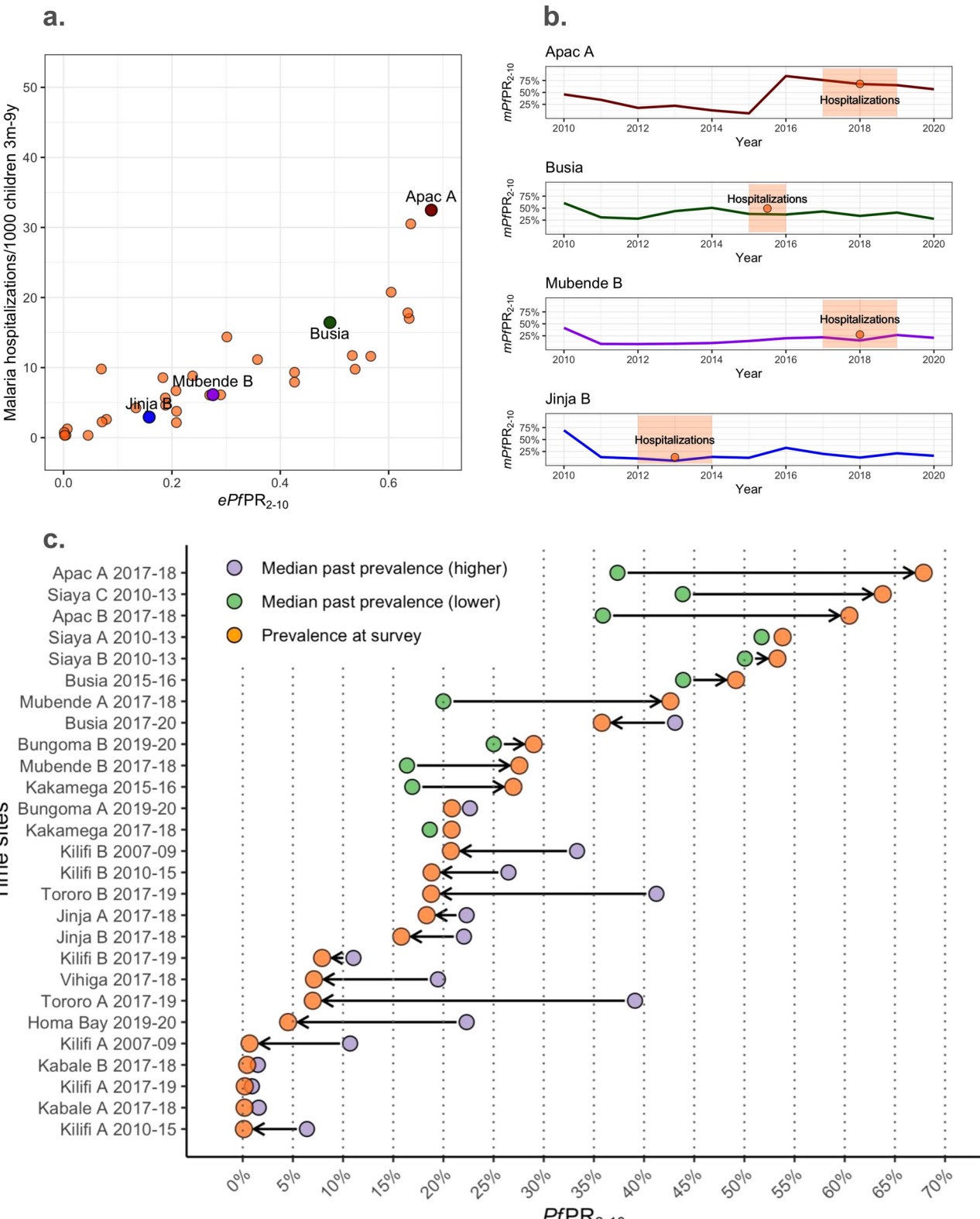

**Fig. 1 | The contemporary relationship between $PfPR_{2-10}$ - malaria hospitalization rates and past exposure. a** The $ePfPR_{2-10}$-severe malaria incidence empirical relationships highlighting four representative time-sites (red, green, purple, and blue colored dots) within all time-sites (orange dots). **b** Present day $mPfPR_{2-10}$ over time in four representative time-sites (red, green, purple, and blue colored lines) and time-site $ePfPR_{2-10}$ (orange) with highlighted time periods for which hospitalization incidence estimates were available for the empirical relations in (**a**). **c** Summarizing the prevalence trends over time estimated for each time-site as an increasing trend -when the estimated median $mPfPR_{2-10}$ in the past 7 to 9 years is lower than the $ePfPR_{2-10}$ at the time of assessment of severe outcomes incidence- or decreasing trend the estimated median $mPfPR_{2-10}$ in the past 7–9 years is lower than the $ePfPR_{2-10}$ at the time of assessment of severe outcomes incidence. The median past $mPfPR_{2-10}$ per time-site is plotted as yellow (for those with increasing trends) or red (for those time-sites with reducing trends), and $ePfPR_{2-10}$ per time-site is plotted in orange. Estimates have been computed among the 27-time sites with at least seven years of available past $mPfPR_{2-10}$ estimates.

### a. Recovery of the historical relationship given steady transmission

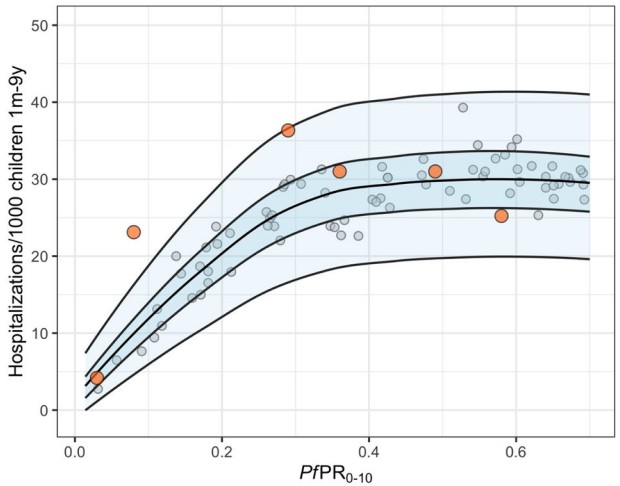

### b. Recovery of the contemporary relationship given steady transmission

### c. Individual base modeled $PfPR_{2-10}$ and hospitalization rates time-series

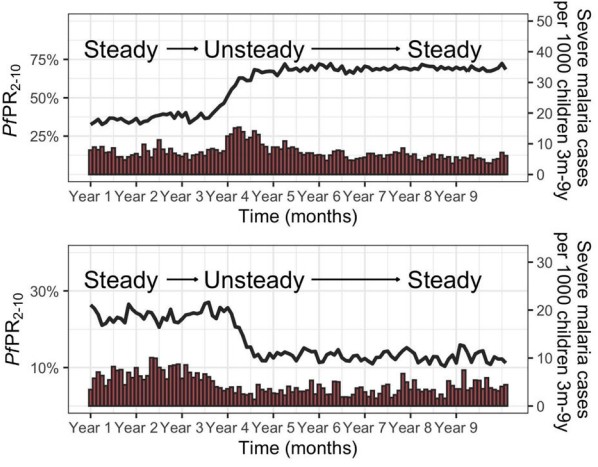

### d. Recovery of the contemporary relationship given unsteady transmission

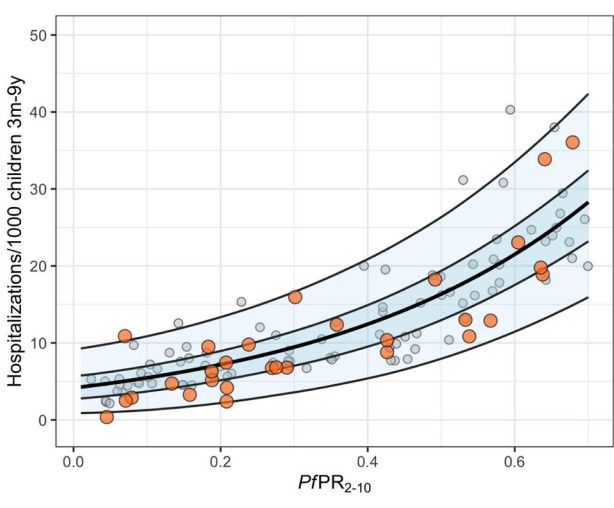

**Fig. 2 | Recovery of the $PfPR_{2-10}$-malaria hospitalization rates relationship.** Showing the empirical $PfPR_{2-10}$-hospitalization rates relationship obtained from **a** the historical dataset (orange dots), and **b** the contemporary dataset (orange dots) overlapping respective $PfPR$-hospitalization model-based estimates obtained through simulations consistent with steady transmission (gray dots, $n = 100$) and respective levels of health care access, treatment and comorbidities (**b**), and the best-fit model-based log-logistic regressions (median black line, blue ribbons 50% and 95% prediction intervals). **c** Representative simulation patterns of $PfPR_{2-10}$

(black lines, increasing at the top and decreasing at the bottom) and subsequent malaria hospitalization incidence over time (red columns). **d** The empirical $PfPR_{2-10}$-hospitalization rates relationship obtained from the contemporary dataset (orange dots) overlapping $PfPR_{2-10}$-severity model-based estimates obtained through simulations scenarios consistent with time-varying exposure (gray dots, $n = 100$) and levels in health care access, treatment, and comorbidities, and the best-fit model-based log-linear regression (median black line, blue ribbons 50% and 95% prediction intervals).

influence the risk of hospitalization. Thus, we further assessed the impact of this variability on malaria exposure on the ongoing hospital admission risk estimated from our individual-based model scenarios. Here, we define steady exposure as the exposure of a specific population that does not change substantially over the years, and we define unsteady exposure as an exposure that shows substantial increasing or decreasing dynamics over time. We performed simulations that included generic step-up and step-down exposure dynamics with differences between pre- and intra-survey prevalence within the range of those observed empirically. Further details are found in the Supplementary Note 3. For each scenario analysis, we performed over 1000 individual-based simulations. We tested the model outputs—$PfPR_{2-10}$ and incidence rate of malaria admissions—against three regression

models, namely an intercept-only, a log-logistic, and a log-linear model, which was previously used to evaluate the prevalence-hospitalization relationship[10].

The historical dataset shows higher levels of hospitalization rates at similar prevalence, consistent with the expected reduction of comorbidities and improved management effectiveness in the contemporary dataset (Fig. 2 and Supplementary Note 5). Figure 2a, b shows (1) the historical (Fig. 2a) and contemporary (Fig. 2b) empirical estimates, (2) their respective time-frame-specific simulations under the assumption of steady exposure, and (3) the regression-based predicted relationship, shown as the median and 50% and 95% prediction intervals. For both the historical and the contemporary predictions, the log-logistic regression model provides a better fit to the

model outputs, with the asymptote reaching around 60% and 40% $Pf$PR$_{2-10}$, respectively. However, while the prediction model based on steady transmission is consistent with the historical empirical data, it fails to recover the contemporary dataset relationship, with a substantial number of time-sites outlying from the predictions range, particularly for the highest $ePf$PR$_{2-10}$ values (i.e., over 60%).

Given that model predictions based on steady exposure do not capture the pattern of the empirical contemporary data, we further evaluated scenarios with unsteady past malaria transmission (either increasing or decreasing dynamics), and how these different trends affect the $Pf$PR$_{2-10}$-hospitalization rate incidence relationship. Specifically, we performed simulations that included step-up and step-down exposure dynamics with differences between pre- and intra-survey prevalence within the range of those observed empirically (see Methods). Simulations captured representative patterns of time-varying exposure, with decreasing, steady or increasing transmission before computing severe disease incidence (Fig. 2c) over the range of $Pf$PR$_{2-10}$ values. The best fit to a regression model is then obtained using the log-linear model (Fig. 2d). Based on performance metrics[16], simulations under the assumption of unsteady patterns of past exposure predict the relationship of the contemporary dataset more accurately than those based on steady exposure. Further, modeled predictions under unsteady exposure show hospitalization rates in different age groups increase towards higher $ePf$PR$_{2-10}$ in a similar way as observed in the empirical estimates (Fig. S2). However, this is opposite to the pattern in historical observations under steady state exposure; severe disease incidence among youngest children is typically higher in high transmission settings than in low transmission settings, whereas the opposite occurs among older children[4,7–9]. Consistent with these results, our model recovers more accurately the observed hospitalization age structure (Fig. S3, left column for representative time-sites) under the unsteady transmission assumption (Fig. S3, middle column) than under the steady-state assumption (Fig. S3, right column). Further, when assessing the model estimates of hospitalization risk later in time (i.e., allowing the scenarios to maintain a steady-state level transmission over 5 years), the prevalence-hospitalization rates relationship transitions to the asymptotic pattern expected for steady transmission (Fig. S4).

## Discussion

Via scenario analysis, we have systematically evaluated the major clinical and epidemiological determinants influencing the occurrence of malaria hospitalization upon infection at the population level and thus proved a contemporary characterization of relationship trends and changes between malaria community prevalence and life-threatening disease risk in children. We found that the asymptotic relationship between prevalence and hospitalization disease risk, expected under a relatively steady transmission, is lost when children have been exposed to unstable, time-varying past malaria transmission. Overall, our analyses support the assumption that substantial fluctuations in malaria transmission over the years have led to a particular prevalence-hospitalization relationship observed among the East-African settings[10], where increasing prevalence does not necessarily lead to saturating disease risk but increases toward the highest rates in an exponential manner. However, our analyses show that if transmission is further maintained at a steady-state level over sufficient time, disease risk would also eventually re-equilibrate back to the asymptotic relationship relative to the parasite rates (fig. S4). To date, inference frameworks aiming to estimate or predict severe outcomes of malaria used for policy decisions and public health action do not explicitly include past exposure as an independent variable[5,17]. Our analysis framework is capable of reconciling historical and contemporary observations encompassing three decades in sub-Saharan Africa and underscores the importance of taking the variability of past

malaria exposure among children into account when predicting severe disease risk.

Our analyses have several limitations that need to be acknowledged. First, we use community-based hospitalization in non-urban settings as an empirical proxy of the incidence of severe outcomes of malaria. While these estimates could under- or over-estimate the true number of severe outcomes, the data was obtained aiming to standardized the under-ascertainment of cases, estimates would be affected by site specific treatment-seeking behaviors and therefore represent a lower limit for hospitalization rate at each time site. Also, the empirical contemporary dataset does not necessarily represent urban settings, where the referral pathways from infection to hospitalization might be more complex to understand or subject to other potential biases. Further, the curated data does not include cases where malaria was not the major syndrome for hospitalization. Our modeling approach allows access to health (i.e., access to diagnosis, treatment, and/or hospitalization) to randomly vary within the range of the rates estimated for the three countries, with sensitivity analysis showing that deviations from this assumption do not influence the overall relationship between prevalence and hospitalization. Nevertheless, if the data represents stronger deviations from these assumptions regarding case identification but remains relatively similar across time sites, the overall prevalence-severe disease trend will still hold. Second, the empirical $Pf$PR estimates obtained through community and school surveys might not necessarily reflect the underlying prevalence dynamics for the full catchment population, given how heterogeneous malaria exposure can be at a very granular spatial level. However, we obtain a similar prevalence-hospitalization relationship using estimates computed using the geospatial model (see Section "Discussion", Fig. S10), thus supporting the assumption of the empirical values being a representative summarizing value for the catchment populations. Also, the time series of $Pf$PR values used to compute exposure steadiness can bear high uncertainty on the precise estimates. Nevertheless, given that we did not aim to replicate the prevalence changes over time but addressed this matter focusing on the relative change, our framework will remain well informed if the estimates approximate true trends. Third, our model OpenMalaria simulations are based on spatially homogeneous malaria transmission because the catchment populations in the empirical data are small. Assuming a heterogeneous transmission structure can affect the magnitude of the clinical outcomes, central estimates will not change[18]. Fourth, while parameterization of the hospitalization rates, efficacy of treatment, and comorbidities are consistent with the literature, for simplicity, we assumed similar ranges of values across time-sites with stochastic variation. In Supplementary Note 4, we provide an uncertainty analysis of these mechanistic parameters, showing that our results hold under no major deviations from tenable assumptions. Last, we have applied parametric regression models to simulation data, which likely misspecifies the mechanical interpretation of the model. Still, we believe this is justified given this approach was originally used to determine the empirical relationship[10] and we aimed to replicate the trends under potential plausible mechanistic scenarios.

We have provided evidence that variation in malaria transmission and subsequent disease protection after life-long exposure can strongly influence severe disease risk estimates under otherwise equivalent ongoing force of transmission. Notably, frequent implementation and withdrawal of infection prevention strategies can strongly contribute to unsteady malaria exposure patterns and thus increase severe disease risk. Other potential sources of variability in the exposure include substantial movement of individuals from areas with different community prevalence (i.e., via increasing or reducing the overall population-level susceptibility to severe disease) or strong environmental changes influencing entomological inoculation rates

such as urbanization or climatic drivers (i.e., prolonged drought or excessive rainfall).

Under constant, similar malaria exposure and health access rates, a population of children with higher immunity will substantially reduce their risk of severe malaria and, therefore, the number of severe cases. However, if malaria transmission has marked changes, either sudden reductions or increases, the resulting severe disease will be significantly lower or higher, respectively, than if the population had remained under constant transmission. This understanding is critical to evaluate the effects of interventions, and such mechanistic processes need to be included in future analytical approaches providing predictions of malaria disease burden. The processes that must be incorporated into disease burden estimates are best defined through differences in the age–structure patterns of the risk conditional to past exposure. For example, it is expected that following a strong reduction in malaria transmission, severe cases among age groups of children at a certain prevalence will be reduced earlier in time but will likely increase in later periods if malaria prevalence reaches steady levels. Similarly, the withdrawal of effective prevention and control strategies will lead to a higher number of severe cases than those expected when malaria prevalence has remained unchanged over time. In short, if past exposure and the dynamics described here are not accounted for in burden estimates, it will lead to long-term overestimation of severe malaria risk in places with recent effective interventions. And conversely, it will lead to long-term underestimation of severe malaria in places with deterioration of interventions. Finally, our findings underpin the need to build back rigorous clinical surveillance of severe malaria under the changing landscape of parasite exposure in Africa. It is striking that only two longitudinal clinical series exist since the launch of the Roll Back Malaria initiative in 2000[19,20].

Overall, our findings provide evidence that inference in malaria epidemiology, such as the generation of counterfactual scenarios with predictions on clinical outcomes for policy decisions, should account from now on past exposure and subsequent protection to avoid substantial bias in such risk predictions and highlight the increase in the complexity of malaria epidemiology arising from unsteady transmission dynamics.

## Methods

### Contemporary malaria hospitalization incidence data and paired community prevalence estimates

The contemporary *P. falciparum* malaria hospitalization incidence data has been obtained from 35 time-sites in Kenya ($n = 18$), Uganda ($n = 14$), and Tanzania ($n = 3$) between 2006 and 2020 and has been previously described elsewhere[10]. The data was collated from individual records of 21 hospitals, including hospitalized malaria cases in children resident in specific catchment areas within a 30 km range and excluding urban settings. Given potential differences in treatment-seeking behaviors at the different sites, the computed estimates represent the lower limit for hospitalization incidence, thus requiring further assumptions for adjusting to under ascertainment (see model parameterization in the Supplementary Text). for In the present analysis, cases were included if malaria was deemed the primary cause of hospitalization. Per each time-site, the number of hospitalizations among children 3 months to 9 years old per 1000 children per year was computed as a proxy of life-threatening disease incidence from the catchment population, obtained from census data and census data projections. Hospitalization data was paired with community surveys performed at the same periods of the hospital surveys for each time-site, adjusting for diagnostic accuracy (i.e., microscopy vs. rapid test) and standardized by computing the $Pf\text{PR}_{2-10}$ as described elsewhere[21,22]. Further details on hospitalization data, community prevalence, and estimates of hospital catchment population can be found in Supplementary Note 1.

### Modeled community prevalence time-series

We assessed the past exposure dynamics using catchment site-specific time series of modeled age- and test-standardized parasite rates estimates, referred to as $mPf\text{PR}_{2-10}$[13,15]. Modeled $mPf\text{PR}_{2-10}$ estimates were explicitly obtained for each time-site catchment population using a geospatial model detailed elsewhere[13,15], a Bayesian hierarchical geostatistical framework based on more than 180000 geo-coded empirical prevalence survey data points from East Africa, interpolated in time to $1 \times 1$ km resolutions using climatic and ecological covariates. For a set of sites ($n = 11$), annual $mPf\text{PR}_{2-10}$ estimates are available since 2000, while for the rest ($n = 15$) available data begin in 2010. Time series of all the available $mPf\text{PR}_{2-10}$ per site are shown in Fig. S1. Thus, for 27 out of the 35 present-day time sites, $mPf\text{PR}_{2-10}$ time series included at least seven-time points of past annual estimates. Further, to estimate the gap between past and present transmission, we first computed the median value of the annual $mPf\text{PR}_{2-10}$ time-series for each of the time-sites, under the assumption that it roughly represents the life-long transmission at which the surveyed population of children have been exposed in the previous years previous. This median $mPf\text{PR}_{2-10}$ can then be compared to the empirical $Pf\text{PR}_{2-10}$ that is estimated at the time of surveying the malaria hospitalization incidence.

### Historical malaria hospitalization incidence data and paired community prevalence estimates

As an alternative source of data, we analyzed a historical dataset obtained between 1992 and 1997, encompassing the relation between severe disease incidence measured as malaria hospitalization rates and the $Pf\text{PR}_{2-10}$ up to 70% and published elsewhere[9]. Similar inclusion and exclusion criteria had been used to obtain both datasets explicitly to allow comparison. Thus, comparable approaches to those used in the contemporary dataset were used to estimate malaria hospitalization incidence at six hospital time-site locations in The Gambia ($n = 3$), Kenya ($n = 2$), and Malawi ($n = 1$) and paired community prevalence estimates[7,9,11]. Further details can be found in Supplementary Note 1.

### Agent-based model of malaria transmission, immunity, and disease dynamics

To recover the relationship between hospitalization rates and $Pf\text{PR}_{2-10}$, we used an individual-based model of *P. falciparum* malaria transmission and disease dynamics, OpenMalaria, previously described and calibrated elsewhere[12]. Briefly, OpenMalaria features within-host parasite dynamics, the progression of clinical disease, development of immunity, individual care-seeking behavior, vector exposure, and pharmaceutical and non-pharmaceutical antimalarial interventions at vector and human level (https://github.com/SwissTPH/openmalaria/wiki)[14,23,24]. The full model is calibrated to fit 23 parameters to 11 objectives representing different epidemiological outcomes, including age-specific prevalence and incidence patterns, age-specific mortality rates, and hospitalization rates, using a Bayesian optimization approach[12]. A detailed description of the model and references for the key technical details are provided in Supplementary Note 2.

### Model parameterization on disease management, comorbidities, and health access

In order to compare model predictions to empirical data, we first parameterized several key inputs of the individual-based model to be consistent with both historical and contemporary datasets and the corresponding existing clinical and epidemiological knowledge, including different sets of variables addressing (a) the individual probability of malaria exposure per time-step to cover a range of $Pf\text{PR}_{2-10}$ up to 70%, (b) the effectiveness in malaria case management, including the rate of access of uncomplicated and severe malaria cases to health-care (i.e., rate of accurate diagnostic of true occurrence and subsequent timely treatment) and the efficacy of the available malaria drugs at each period (e.g., the efficacy of artemisinin derivatives

combination therapies in clearing malaria), and (c) the co-occurrence of other diseases with influence on malaria that affect the severe progression of malaria across age-groups.

To evaluate the contemporary dataset, model parameterization for the main analysis was as follows: (1) $PfPR_{2-10}$ estimates ranged up to 70%; (2) drug efficacy ranged between 85 and 100%[25,26]; (3) the rates of individuals suffering from malaria that required hospitalization who received diagnostic and treatment ranged from 60 to 90% while the rate of individuals with uncomplicated malaria accessing diagnostic and treatment ranged 20–60%[27]; and (4) co-occurrence of diseases contributing to malaria hospitalization was substantially reduced by 60–80% since the 1990s[28–30]. For (2–4), we parameterized the model by randomly sampling values from a uniform distribution defined by the respective ranges. Sensitivity analysis for these assumptions can be found in the Supplementary Note 3. To evaluate the historical dataset, we include the following: (1) $PfPR_{2-10}$ estimates ranged up to 70% (2) first-line drug efficacy for uncomplicated malaria was negligible[31]; (3) access rates of malaria requiring hospitalization ranged from 40–60%[8,32], and (4) co-occurrence of diseases contributing to malaria hospitalization approximated a hyperbolic decay distribution over age-groups[8]. Details on key model parameters and submodels are provided in Supplementary Note 3.

### Time-varying malaria exposure on severity estimates
To evaluate how time-varying malaria exposure rates influence the ongoing $PfPR_{2-10}$-hospitalization rates relationship, we produced a set of plausible malaria-transmission scenarios consistent with time-varying exposure as for those obtained from a Bayesian hierarchical geostatistical framework[13,15] across all our time-sites. Particularly, we evaluated how a rapid reduction or increase of the $PfPR_{2-10}$ level over 6 months (e.g., $PfPR_{2-10}$ increasing from 10% to 70%) could affect the severe malaria incidence relative to a steady-state transmission assumption described before. This was performed by setting (1) the initial exposure rate (i.e., the individual probability of exposure to infectious bites pre-survey as a single rate over simulation time-steps), independent from (2) the final exposure rate (i.e., the probability of exposure during the survey period, also as a single rate over time steps). For simplicity, we set constant initial and final exposure rates. The relationship between the initial and final exposure rate was parameterized to reflect archetypical $mPfPR_{2-10}$ patterns,–depicted in Fig. 1b–showing (1) a substantial increase in the $mPfPR_{2-10}$ (e.g., Apac A), (2) relative constant $mPfPR_{2-10}$ (e.g., Busia), and (3) substantial decrease in the $PfPR_{2-10}$ (e.g., Jinja B). Thus, our definition of time-varying (unsteady) exposure comprises both substantial increasing and decreasing dynamics. For each time site with retrospective data encompassing at least 7 years, we computed the average $mPfPR_{2-10}$ available up to 9 years prior to the date when the hospitalization data was available, used it as a proxy of the initial exposure rate, and computed the fold-change $PfPR_{2-10}$. We then performed a local polynomial regression model to obtain predictions of the relationship between the final exposure rate and the corresponding relative change per time site. See fig. S5 in Supplementary depicting the relationship between the contemporary community prevalence as $ePfPR_{2-10}$, and estimated relative change at survey computed from the median value of $mPfPR_{2-10}$ of the past 7–9 years for each time-site. We then used these inputs of exposure to produce simulations using a combination of initial and final exposure rates to map a range of simulation-based $PfPR_{2-10}$-hospitalization rates.

### Scenario analysis to recover the empirical relationship
To evaluate under which scenarios OpenMalaria can recover the empirical prevalence-hospitalization, we implemented an iterative 4-step procedure to explore hypotheses of the impact of the determinants on disease risk and changes in disease risk–see sections above for details. The procedure is represented schematically in the

Supplementary (Fig. S6). For each scenario analysis, we performed four iterative steps as follows:

1. Using a high-performance computing framework (http://scicore.unibas.ch/) we performed 1000 population-level individual-based model simulations of malaria transmission at a steady-state (i.e., same entomological inoculation rate for each simulation) over long periods of time (i.e., over 90 years which ensures that lifelong malaria exposure has occurred all the generations evaluated prospectively); computing hospitalization rates among children 3 months- 9 years per 1000 persons-year across values of $PfPR_{2-10}$ within the input range. In each of the simulations, we parameterized the model, mapping the range of values set up for the major epidemiological determinants described in the previous section, namely disease management (which includes rates and efficacy of diagnostic and treatment of severe and uncomplicated malaria) and co-occurrence of comorbidities. We set the parameters according to the empirical dataset we were evaluating (i.e., historical or contemporary). At the same time, we evaluated the model outcomes simulating steady and unsteady transmission.

2. In order to assess how accurately the model simulations represent the empirical data, we evaluated the performance of two regression models to recover the prevalence-hospitalization relationship obtained via scenario analysis, namely a log-linear model and a log-logistic model. The approach was chosen to be consistent with the previous analysis framework of the contemporary dataset[10], which analyses severe malaria syndrome-specific cases against prevalence. See Supplementary Note 3 for details on the regression models, model selection criteria, and computation of uncertainty estimates.

3. We used the best-fit regression model among the above to evaluate how the modeling framework predicted the empirical $ePfPR_{2-10}$-hospitalization incidence relationship. Specifically, we evaluated the accuracy of the model predictions using prediction performance metrics[16] (see Supplementary Note 3 and Table S2).

4. We updated model assumptions based on the evaluation above and reparametrized the scenario(/s) accordingly.

### Evaluation of key assumptions for scenario modeling
To evaluate the robustness of the simulation-based predictions of the $PfPR_{2-10}$-hospitalization relationship under time-varying exposure, we performed three major sensitivity analyses: (a) how increased or reduced rates of both uncomplicated case management and hospitalizations at the population level influence the severe disease incidence and prevalence relationship; (b) how the relationship changes over higher or lower drug efficacy estimates; (c) how changes on the incidence of comorbidities influence the severe disease risk. Details of the evaluation are provided in Supplementary Note 4 and Figs. S7 and S8.

### The age structure of the severity estimates and the effect of time-varying malaria
Last, to assess if the model framework recovers hospitalization risk over different age groups, we compared both empirical and simulation-based estimates disaggregated by age. We performed the 4-steps iterative analysis described above for the contemporary dataset under the steady and time-varying transmission assumptions and evaluated the $PfPR_{2-10}$-hospitalization relationship by age. More specifically, we computed hospitalization incidence over age groups at values of $PfPR2-10$ equal to those estimated in the for four representative time sites Apac A, Busia, Mubende B, and Jinja B. Further details on the age-structured analysis are provided in Supplementary Note 5.

### Reporting summary
Further information on research design is available in the Nature Portfolio Reporting Summary linked to this article.

## Data availability

Empirical contemporary data used in this analysis have been curated and uploaded to the Harvard Dataverse: https://dataverse.harvard.edu/dataset.xhtml?persistentId=doi:10.7910/DVN/XGDB3K. Correspondence and requests for materials should be addressed to the KEMRI Wellcome Data Governance Committee (dgc@kemri-wellcome.org). These data are available through a formal requesting process to the KEMRI Institutional Data Access/Ethics Committee. Guideline details can be found on the KEMRI Wellcome website: https://kemri-wellcome.org/about-us/#ChildVerticalTab_15.

## Code availability

Model code, plotting code, and simulation data are available at https://github.com/PDeSalazarSwissTPH/SevereMalaria.

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

## Acknowledgements

Calculations were performed at sciCORE (http://scicore.unibas.ch/) scientific computing center at the University of Basel. *Funding*: This study was partly funded by Bill & Melinda Gates Foundation (INV-025569 to MAP) supporting M.A.P. and P.M.S. M.A.P. also received support via a Swiss National Science Foundation Professorship (PP00P3_203450 to M.A.P.); R.W.S is supported under a Wellcome Trust Principal Research Fellowship (212176/Z/18/Z), which also provided support for A.K.; R.W.S., A.K. and S.A. are grateful for the support from the Wellcome Trust to the Core Award for the East Africa Major Overseas Program (203077/Z/16/Z). S.A. was supported by the CDC Foundation through funding from the World Health Organization for the RTS,S evaluation (project requisition number 2018/854999).

## Author contributions

Conceptualization: P.M.S., M.A.P., A.K., R.W.S.; methodology: P.M.S., M.A.P., R.W.S.; investigation: P.M.S., S.A., A.M., A.K., A.C., R.W.S., M.A.P.; visualization: P.M.S., M.A.P.; funding acquisition: M.A.P., R.W.S.; writing —original draft: P.M.S., M.A.P., R.W.S.; writing—review & editing: P.M.S., M.A.P., R.W.S., A.K., A.C., S.A., A.M.

## Competing interests

The authors declare no competing interests.

## Ethics declaration

Kenya hospital surveillance: Kenya Medical Research Institute/ Scientific and Ethics Review Unit (KEMRI SERU) IRB numbers 1433, 3057, 3149; 1801, 2558, 2465, 3459, and 3771; Uganda hospital surveillance approved by the CDC as public health surveillance/non-research (NCEZID 031416), Tanzania hospital surveillance approved by the ethics committees of the National Institute for Medical Research, Tanzania, and the London School of Hygiene and Tropical Medicine. Kenya community parasite surveys KEMRI SERU IRB numbers 3149, 3592, 1801, 2558, 2675, 2801, and 3822; Uganda community prevalence surveys Vector Control Division Research Ethics Committee, MoH, Kampala: VCDREC/117.
