## [Peer Review File · Nature Communications]

Severe outcomes of malaria in children under time-varying exposureReviewers' Comments:

Reviewer #1:

Remarks to the Author:

The authors set out to understand severe malaria risk under time-varying exposure using hospitalization data paired with prevalence data from the same catchment area. The authors use scenario analysis to show that the relationship between prevalence and hospitalization risk breaks down when past malaria exposure occurs within an unsteady transmission state. I agree with the authors conclusion that inferences in malaria epidemiology moving forward need to account for past exposure dynamics, but that does not seem to me to be a particularly novel finding. Additionally, I have a number of questions that I believe the authors should consider.

The authors rely on hospitalization data and prevalence estimates from within catchments. However, individuals (particularly mothers concerned for their sick children) do not necessarily adhere to the health system referral hierarchy. For example, it is known that individuals may seek the highest standard of care available to them, regardless of whether or not that care can be found within a given catchment area. This is a critical limitation. Another critical limitation is that they exclude urban settings, which are major centers of healthcare seeking behavior (i.e. rural-urban movement to seek health care). This is in fact common in my field sites. Do the authors have data on whether or not these catchment areas reflect actual treatment-seeking behavior, especially for severe malaria (where travel to an urban center may be more likely)? Or, failing that, do they have any sensitivity analyses under different assumptions (e.g. XX-YY% seek care elsewhere and the effects on estimates)?

Related to the above, another critical limitation is that the authors note that a catchment area was defined using a 30km radius around a hospital, and state that this "maximizes access to health" in line 236 of the SI. I'm skeptical of this threshold, particularly as it relates to hospitalization rates for malaria. In my study sites, a hospital is frequently more than 30km away. Is there any literature to suggest that 30km maximizes access to health? If so, that would be an impactful finding in itself. At present, this feels arbitrary in a way that would require sensitivity analyses that considered both different radii and different ways of measuring distance (e.g. such as distance along a road).

For each time-site, the authors pair the average number of hospitalizations due to malaria among 3mo-9 year olds (per 1000 children/year) with standardized prevalence estimates from the same catchment area as the hospital. These data were taken from community and school-surveys undertaken during the hospital surveillance period. Given the focal nature of malaria transmission due to mosquitos' limited mobility and variability in land cover and other environmental features over a 30km radius, it seems entirely plausible that the PfPR surfaces don't reflect the "true" underlying risk of the patients who are hospitalized. This would, it seems, affect the main findings since the pairing wouldn't be doing what it is designed to do. The authors don't even provide a map of the study communities in the manuscript to compare where hospitalized patients reside relative to the PfPR survey communities.

A couple minor comments:

Figure 1C is nearly impossible to read

The description of the mPfPR is somewhat unclear. The authors note that this is a modeled estimate, as opposed to the empirical estimate (ePfPR). The authors then refer to this modeled estimate as median PfPR. Are they taking the median of the posterior from the model?

Reviewer #2:

Remarks to the Author:

This is a very interesting paper looking at the relationship between (time-varying) malaria exposure and severe disease risk. The authors re-examine an established contemporary dataset of malaria hospitalizations in children as well as historical data on hospitalizations, in conjunction with geospatial estimates of time-varying malaria prevalence and a mathematical model of malaria transmission. The authors show that discrepancies between the contemporary and historical datasets with respect to the shape of the severe incidence – prevalence relationship can be explained by recent changes in malaria exposure, whether that is a recent decrease or recent increase.

The authors have clearly thought deeply about the empirical observations, and I appreciate that the authors who originally published the contemporary dataset have continued to examine this data in detail to see what else can be learned from it. The result on steady vs unsteady transmission is, in retrospect, perhaps rather obvious but the authors have done a nice job providing strong support for it.

My main comment on this manuscript is that I found parts quite confusing and not clearly motivated, particularly in relation to the use of the mathematical model.

Minor comments:

L23-4: is there a difference between severe malaria and hospitalizations? It would be helpful if this work used one term exclusively, unless some aspects of the analysis were on severe malaria and some were on hospitalizations.

L27: perhaps helpful to be clear that “unsteady” is in either direction (increasing or decreasing)

L43-4: is the implication that inference accuracy depends only on understanding how clinical outcomes are related to exposure and not to other factors such as access to high-quality treatment?

Fig 1a: should x axis label be ePfPR rather than mPfPR?

Fig 1b: consider indicating the ePfPR on these plots

L115: the substantial decrease in Jinja B is based only on the 2010 mPfPR, is that correct? Why did some sites have mPfPR going further back historically than others? Do we think the 2010 estimate reflects multiple years of high transmission before 2010, or could it be an outlier?

L126-7: It would be helpful to motivate a little the approach of using the model to assess the impact of changing transmission. The current presentation seems to suggest the model is magic --- but I suspect there are good reasons to believe the model is capturing something (changing population immunity in response to changing exposure, and resulting changes in clinical presentation) accurately. Why would we think that? And therefore why would we consider using a model like that to better understand this severe dataset?

L181: what are the representative patterns? This is clearer after reading the supplement, but I suggest also clarifying here whether you are simulating a generic step-up or step-down, or if the change is meant to be site-specific.

L272: sentence beginning “When the population immunity” I did not understand this at all

L349: “access rates” this refers to access to treatment? Perhaps re-word this sentence

Section beginning L347: were simulations / scenarios drawn from these options (parameters sampled from these ranges)? Or were they deliberately constructed? Based on the end of Section S4 I think they were deliberately constructed based on DHS and other data, is that right? Unclear from this

section.

L393: out of curiosity...why is it necessary to simulate 90 years if you only look at children under 10? Why not simulate just 10 years?

Supplement:

L193-8: totally completely confused by this. Perhaps more specificity would be helpful? I'm not even sure what to ask that would help me understand.

L200: what does it mean to have a predictor "population-level immunity"?

L251 section and Fig S2 and S3: this part also not clear to me. It's not easy to eyeball the figures and see what the authors are talking about here with age patterns of severe incidence by PfPR.

Fig S6: this figure was quite helpful to understand what the authors were doing. I would consider putting it in the main text if there is room, and perhaps also putting more detail on it (what assumptions are being used to parameterize the IBM? For example), and fitting it into the larger conceptual flowchart of the analysis approach.

Response to reviewers

Reviewer #1 (Remarks to the Author):

1. The authors set out to understand severe malaria risk under time-varying exposure using hospitalization data paired with prevalence data from the same catchment area. The authors use scenario analysis to show that the relationship between prevalence and hospitalization risk breaks down when past malaria exposure occurs within an unsteady transmission state. I agree with the authors conclusion that inferences in malaria epidemiology moving forward need to account for past exposure dynamics, but that does not seem to me to be a particularly novel finding.

Thank you for the review and thoughtful critique below. We agree the influence of immunity in malaria outcomes is well established as shown by pathogenesis, immunological and clinical studies, however, the quantification of the same at the population level remains relatively understudied. Subsequently, epidemiological studies rarely (if ever) address quantitatively the influence of previous exposure and elicited immunity on malaria disease outcomes. Significantly, while the influence of immunity is well known on disease outcomes, delay or abrupt changes to immunity or exposure risk are currently **not** considered in the WHO (or other) frameworks for global burden of disease estimates produced yearly. Sufficiently granular data on exposure (for time and space) is not easily available as empirical observations would require high resources (thus the existence of estimates based on models). As a result, to our knowledge, generalizable modeling frameworks essential for estimating disease burden or the effect of interventions do not explicitly include past, life-long transmission as a quantitative independent variable of severe outcomes of disease. We have now included the following in the Discussion section for clarity:

“To date, inference frameworks aiming to estimate or predict severe outcomes of malaria used for policy decisions and public health action do not explicitly include past exposure as an independent variable^{1,2}. Our analysis framework is capable of reconciling historical and contemporary observations encompassing three decades in sub-Saharan Africa and underscores the importance of taking the variability of past malaria exposure among children into account when predicting severe disease risk.”

Additionally, I have a number of questions that I believe the authors should consider.

2. The authors rely on hospitalization data and prevalence estimates from within catchments. However, individuals (particularly mothers concerned for their sick children) do not necessarily adhere to the health system referral hierarchy. For example, it is known that individuals may seek the highest standard of care available to them, regardless of whether or not that care can be found within a given catchment area. This is a critical limitation.

We agree with the reviewer that estimating accurately true hospitalization rates from the community from hospitals is challenged by critical limitations. Because of that, datasets with sufficient quality to inform epidemiological analysis are extremely scarce. The two datasets included here are among the very few available to date - together with the clinical series from Manhica, Mozambique ³. Nevertheless, information obtained from analyzing empirical data is very valuable, particularly when tackling questions for which data is difficult to obtain due to

logistics and cost limitations. Here, we do not claim that our estimates are true values, but rather reasonable representations -or proxies- that allow us to evaluate different time-sites in a standardized way. The interpretation in our work is that the hospitalization estimates represent a minimum value of hospitalizations. Because the analyses here rely on the relationship between prevalence and hospitalization rates, our conclusions will hold if the ascertainment bias (in this case, what proportion of cases are missed that otherwise should have been hospitalized if attending the hospital) is relatively homogeneous across the time-sites. Acknowledging this, we have addressed this limitation in our modeling strategy, by allowing hospitalization rates to vary between 60-90%, which are values consistent with DHS estimates for the three countries. Also, in our sensitivity analysis we show that higher (over 90%) or lower rates (between 40-60%) will modify the shape of the pattern by increasing or decreasing the overall level, but do not explain the exponential vs asymptotic pattern.

We have now clarified this limitation in the Discussion section as follows:

“Our modeling approach allows access to health (i.e., access to diagnosis, treatment, and/or hospitalization) to randomly vary within the range of the rates estimated for the three countries, with sensitivity analysis showing that deviations from this assumption does not influence the overall relationship between prevalence and hospitalization. Nevertheless, if the data represents stronger deviations of these assumptions regarding case identification but remain relatively similar across time-sites, the overall prevalence-severe disease trend will still hold.”

We also refer to the sensitivity of this particular assumption (access to treatment and diagnostic and referral rates to hospitalization among children with life-threatening malaria in Supplementary, Section S4 and in the Supplementary Fig. S7 under the unsteady state past transmission (top row below) and steady-state past transmission (bottom row below):

“We evaluated the following: A) main scenario: access rates of malaria requiring hospitalization have ranged from 60%-90% while the access rates of uncomplicated malaria treatment ranged 20%-40% based on the higher estimates from the Demographic and Health Surveys program ⁴; secondary scenario 1: health care access rates have been substantially lower, with hospitalizations being between 40% and 60% and uncomplicated malaria treatment rates of 20%-40%, similar to those estimates in the 1990s; secondary scenario 2: access rates were 90-98%.”

“Overall, the different set of parameterizations aimed to be consistent with the following characteristics of the contemporary empirical dataset: a) malaria population-based hospitalization estimates were aligned with DHS estimates ⁴, and b) individuals with primarily reported comorbidities that were not malaria had been excluded in their majority from the analysis dataset.”

Hospitalization rates 60-90%

Hospitalization 40-60%

Hospitalization 90-98%

Results of the sensitivity analysis of our main model parameterization assumptions. a-c) Sensitivity to access to hospital care for severe disease.

3. Another critical limitation is that they exclude urban settings, which are major centers of healthcare seeking behavior (i.e. rural-urban movement to seek health care). This is in fact common in my field sites.

We agree that data does not capture urban settings. We have included the following statements in the limitations, Discussion Section:

“While these estimates could under- or over-estimate the true number of severe outcomes, the data was obtained aiming to standardized the under-ascertainment of cases, estimates would be affected by site specific treatment-seeking behaviors and therefore represent a lower limit for hospitalization rate at each time site. Also, the empirical contemporary dataset does not necessarily represent urban settings, where the referral pathways from infection to hospitalization might be more complex to understand or subject to other potential bias.”

4. Do the authors have data on whether or not these catchment areas reflect actual treatment-seeking behavior, especially for severe malaria (where travel to an urban center may be more likely)? Or, failing that, do they have any sensitivity analyses under different assumptions (e.g. XX-YY% seek care elsewhere and the effects on estimates)?

Thank you for these important questions. The catchment areas were defined to be standardized across time settings but avoid competition with other facilities offering admission, rather than reflect treatment-seeking behaviors. The catchment was also designed to include sufficient

population to avoid variability arising from stochasticity due to too low numbers of hospitalizations. We agree with the reviewer that setting up catchment populations within administrative boundaries (such as county) with a relatively similar radius from a hospital does not guarantee standard treatment-seeking behavior. We do not aim to claim that our empirical estimates reflect the true hospitalization rates, but a lower limit of the incidence of hospitalizations for a given time-site. In other words, while we can not ascertain the true number, we know that at least those cases occur and represent the minimum risk. Further, we have based our model main parameterization to capture different rates of diagnostic/treatment obtained from the DHS as explained above (60%-90%), and provided sensitivity from deviations from this assumption (lower rates of 40-60% and higher rates >90%). For sensitivity analysis please refer to the comment above (query number 2).

We have now clarified the definition and use of the catchment population in the Results section:

“Malaria admissions were assembled from individual records of 21 hospitals representing 35 time-sites in Kenya, Uganda and Tanzania, among children resident in specific catchment areas -within a defined distance radius from the hospital where surveillance took place- excluding urban settings where it was possible to estimate single-year censused population estimates. We assume that the hospitalization rates per time-site represent a lower limit of the hospitalization incidence and can be reasonably comparable when further adjusting for case ascertainment.”

As well as in the limitations of the Discussion section:

“First, we use community-based hospitalization in non-urban settings as an empirical proxy of the incidence of severe outcomes of malaria. While these estimates could under- or over-estimate the true number of severe outcomes, the data was obtained aiming to standardized the under-ascertainment of cases, estimates would be affected by site specific treatment-seeking behaviors and therefore represent a lower limit for hospitalization rate at each time site. Also, the empirical contemporary dataset does not necessarily represent urban settings, where the referral pathways from infection to hospitalization might be more complex to understand or subject to other potential bias. Further, the curated data does not include cases where malaria was not the major syndrome for hospitalization. Our modeling approach allows access to health (i.e., access to diagnosis, treatment, and/or hospitalization) to randomly vary within the range of the rates estimated for the three countries, with sensitivity analysis showing that deviations from this assumption does not influence the overall relationship between prevalence and hospitalization. Nevertheless, if the data represents stronger deviations of these assumptions regarding case identification but remain relatively similar across time-sites, the overall prevalence-severe disease trend will still hold.”

In the Methods Section:

“The data was collated from individual records of 21 hospitals, including hospitalized malaria cases in children resident in specific catchment areas, within a 30 km range and excluding urban settings. Given potential differences in treatment-seeking behaviors at the different sites, the computed estimates represent the lower limit for hospitalization incidence, thus requiring further

assumptions for adjusting to underascertainment (see model parameterization in the Supplementary, Section S1)”

And the supplementary

“Residential data for each admission was matched to the smallest possible area, defined using national census, located within 30 km of the hospital but excluding urban areas. The definition of the catchment population aimed to avoid competition with other facilities, allow computation from available census-data granularity, standardize across sites within the contemporary dataset and allow comparability with the historical dataset, and reduce the case underascertainment within the “whole” hospital catchment population under the assumption that larger distances implied roughly more missing of cases. Population counts among the selected catchments were derived from the most contemporary national census and projected forwards or backwards using district level intercensal growth rates. Age-structures of each population were corrected to single year age groups 3-11 months to 9 years by applying rural household age structures provided for the nearest time-regional matched demographic household survey data. Age specific person years of observation were adjusted for the months of observation included in each temporal series. The exceptions were three sites in Kilifi (Kenya) where actual household continuous population surveillance data were used to define age specific person years of observation. Catchments were selected to avoid competition with other facilities offering admission.”

5. Related to the above, another critical limitation is that the authors note that a catchment area was defined using a 30km radius around a hospital, and state that this “maximizes access to health” in line 236 of the SI. I’m skeptical of this threshold, particularly as it relates to hospitalization rates for malaria. In my study sites, a hospital is frequently more than 30km away. Is there any literature to suggest that 30km maximizes access to health? If so, that would be an impactful finding in itself. At present, this feels arbitrary in a way that would require sensitivity analyses that considered both different radii and different ways of measuring distance (e.g. such as distance along a road).

The definition of the catchment population aimed to avoid competition with other facilities, allow computation from available census-data granularity, standardize across sites within the contemporary dataset and allow comparability with the historical dataset, and reduce the case underascertainment with the whole hospital catchment population under the assumption that larger distances implied roughly more missing of cases. Still, we assumed that the estimated incidence represents the lower limit of the groundtruth, and therefore we adjusted our modeling estimates to underascertainment based on the existing evidence for the countries. Furthermore, we provide with sensitivity analyses for this assumptions, showing that, while deviations change the overall level of the relationship, it does not significantly changes the relationship between prevalence and hospitalization rates (please refer to answers to queries 2 and 4) We understand that this was not well adequately addressed in the text, particularly in the Supplementary, Section S1 and therefore we have now the following:

“Residential data for each admission was matched to the smallest possible area, defined using national census, located within 30 km of the hospital but excluding urban areas. The definition of

the catchment population aimed to avoid competition with other facilities, allow computation from available census-data granularity, standardize across sites within the contemporary dataset and allow comparability with the historical dataset, and reduce the case underascertainment within the “whole” hospital catchment population under the assumption that larger distances implied roughly more missing of cases.”

6. For each time-site, the authors pair the average number of hospitalizations due to malaria among 3mo-9 year olds (per 1000 children/year) with standardized prevalence estimates from the same catchment area as the hospital. These data were taken from community and school-surveys undertaken during the hospital surveillance period. Given the focal nature of malaria transmission due to mosquitos' limited mobility and variability in land cover and other environmental features over a 30km radius, it seems entirely plausible that the PfPR surfaces don't reflect the “true” underlying risk of the patients who are hospitalized. This would, it seems, affect the main findings since the pairing wouldn't be doing what it is designed to do. The authors don't even provide a map of the study communities in the manuscript to compare where hospitalized patients reside relative to the PfPR survey communities.

Thank you. As noted by the reviewer, for the main analysis, we have used prevalence estimates obtained from empirical observations for consistency with the narrative of the paper, which reevaluates data already published at Paton et al, Science 2021⁵. We agree with the reviewer that cross-sectional prevalence estimates might not accurately reflect the complexity of malaria prevalence across space, which can vary even at a household level^{6,7}. However, here we assume that for the specific catchment population, these estimates reasonably summarize the average prevalence as they can be considered as a random sample of different prevalence and therefore serve as a central value. For example, empirical estimates obtained from school surveys will reflect a summarizing prevalence across the different space that covers the households where children live, rather than the prevalence at the school. Similarly, community surveys will reflect a central value of the prevalences across the households of the individuals. For testing the robustness of this assumption, we have now included a paragraph in Section 3 of the Supplementary where we compare the prevalence estimates using empirical data against those obtained through the spatial bayesian modeling framework described in ^{8,9}, used previously to ascertain past exposure for the specific catchment populations. As seen in the new supplementary figure below (fig. S10), the modeled estimates, computed for the time period when the hospitalization estimates were obtained, reproduce the overall “exponential” relationship with only one time-site showing a significant deviation (Fig S10 A). Consistently the correlation between the *m*PfPR and the *e*PfPR shows a linear relationship (Fig S10 B). Although this still does not guarantee that the estimates are true representations of the real parameters, obtaining similar overall patterns using two different sources of information can be seen as internal validation of the model.

*“We also performed a sensitivity evaluation of the empirical prevalence estimates by time-site by comparing the estimates to those obtained using a geospatial model. Modeled *m*PfPR₂₋₁₀ estimates were explicitly obtained for each time-site catchment population using a geospatial model detailed elsewhere ^{8,9}, a Bayesian hierarchical geostatistical framework based on more than 180000 geo-coded empirical prevalence survey data points from East Africa, interpolated in time to 1 x 1 km resolutions using climatic and ecological covariates. As seen in fig. S10, the modeled estimates, computed for the time period when the hospitalization estimates were*

obtained, reproduce the overall “exponential” relationship with only one time-site showing a significant deviation (fig. S10 A). Consistently the correlation between the mPfPR and the ePfPR shows a linear relationship (fig. S10 B).”

Fig S10

Fig. S10. (A) Comparison of the prevalence -hospitalization relationship obtained using the empirical PfPR₂₋₁₀ (orange) and the modeled PfPR₂₋₁₀ (red) computed for the specific catchment populations and time-periods of the hospitalization data. (B) Correlation between the empirical and the modeled PfPR₂₋₁₀ for the same time-sites (gray) and linear relationship (black line)

Nevertheless, we have now noted this limitation in the Discussion Section

“Second, the empirical PfPR estimates, obtained through community and school surveys, might not necessarily reflect the underlying prevalence dynamics for the full catchment population given how heterogeneous malaria exposure can be at a very granular spatial level. However, we obtain a similar prevalence-hospitalization relationship using estimates computed using the geospatial model (see Section 3, fig S10), thus supporting the assumption of the empirical values being a representative summarizing value for the catchment populations. Also, the longitudinal PfPR values used to compute exposure steadiness can bear high uncertainty on the precise estimates. Nevertheless, given that we did not aim to replicate the prevalence changes over time but addressed this matter focusing on the relative change, our framework will remain well informed if the estimates approximate true trends.”

Also, as per advice of the reviewer, we have now included a map that shows the catchment populations and the hospitals where the data was obtained, as well as a summary table (Table S4) describing the sources of the prevalence estimates.

Fig S11

Fig S11. Map plotting all the sites used in the contemporary dataset, zooming in for each of the catchment populations and their relative distance to the hospital where inpatient data was obtained. The figure is reproduced from Paton et al, 2021⁵

Table S4. Sourced from Paton et al. (2021)⁵

Site, dates	Dates Community (CS) or School Survey (SS)*	Positive/examined [Age range, years]	PfPR₂₋₁₀ % [95% CI]**	Citation
Kilifi A, Kenya 2007-09 2010-15 2017-19	2007-09 (CS) 2010-15 (CS) 2017-19 (CS)	10/1645 [0.5-14.9] 1/1439 [0.5-14.9] 0/355 [0.5-14.9]	0.68 [0.32, 1.12] 0.12 [0, 0.34] 0.2 [0, 0.87]	10,11
Kilifi B, Kenya 2007-09 2010-15 2018-19	2007-09 (CS) 2010-15 (CS) 2017-19 (CS)	406/2061 [0.5-14.9] 404/2261 [0.5-14.9] 62/841 [0.5-14.9]	20.75 [18.96, 22.55] 18.82 [17.2, 20.5] 7.91 [6.09, 9.84]	10,11
Kilifi C, Kenya 2018-19	2018-19 (CS)	267/1336 [0.5-14.9]	15.78 [13.9, 17.66]	12
Siaya A, Kenya 2010-13	2010-13 (CS)	234/454 [0.1-14.9]	53.81 [49.12, 58.65]	12
Siaya B, Kenya 2010-13	2010-13 (CS)	588/1155 [0.1-14.9]	53.29 [50.3, 56.33]	13
Siaya C, Kenya 2010-13	2010-13 (CS)	492/808 [0.1-14.9]	63.79 [60.26, 67.2]	13

Busia, Kenya 2015-16 2017-20	2014 (SS) 2019 (SS)	330/596 [4-14.9] 285/681 [4-14.9]	49.16 [45.01, 53.28] 35.79 [32.22, 39.34]	5
Kakamega, Kenya 2015-16 2017-18	2014 (SS) 2018-19 (SS)	65/198 [4-14.9] 204/789 [4-14.9]	26.97 [21.25, 33.09] 20.82 [18.14, 23.59]	5
Vihiga, Kenya 2017-18	2018-19 (SS)	56/596 [4-14.9]	7.1 [5.3, 9.09]	5
Bungoma A, Kenya 2019-20	2019 (SS)	77/297 [4-14.9]	20.85 [16.63, 25.36]	5
Bungoma B, Kenya 2019-20	2019 (SS)	137/392 [4-14.9]	29 [24.69, 33.4]	5
Homa Bay, Kenya 2019-20	2019 (SS)	24/397 [4-14.9]	4.52 [2.81, 6.44]	5
Jinja A, Uganda 2017-18	2019 (SS)	68/400 [5-16.9]	18.33 [14.46, 22.29]	14
Jinja B, Uganda 2012-13 2017-18	2012-13 (CS) 2019 (SS)	80/637 [0.5-14.9] 59/400 [5-16.9]	13.36 [10.8, 16.18] 15.8 [12.26, 19.64]	14, 15
Tororo A, Uganda 2012-13 2017-19	2011-13 (CS) 2019 (SS)	501/820 [0.1-14.9] 27/425 [5-16.9]	64.08 [60.7, 67.63] 6.98 [4.64, 9.64]	14, 15
Tororo B, Uganda 2012-13 2017-19	2011-13 (CS) 2019 (SS)	303/499 [0.1-14.9] 70/399 [5-16.9]	63.54 [59.05, 67.94] 18.78 [14.93, 22.86]	14, 15
Tororo C, Uganda 2012-13	2011-13 (CS)	994/1836 [0.1-14.9]	56.7 [54.29, 59.05]	16
Apac A, Uganda 2017-18	2019 (SS)	127/199 [5-16.9]	67.87 [60.75, 74.95]	17
Apac B, Uganda 2017-18	2019 (SS)	167/294 [5-16.9]	60.46 [54.34, 66.4]	17
Mubende A, Uganda 2017-18	2019 (SS)	77/195 [5-16.9]	42.63 [35.56, 49.9]	17
Mubende B, Uganda 2017-18	2019 (SS)	53/208 [5-16.9]	27.57 [21.56, 33.99]	17
Kabale A, Uganda 2017-18	2019 (SS)	0/400 [5-16.9]	0.18 [0, 0.79]	17

Kabale B, Uganda 2017-18	2019 (SS)	1/400 [5-16.9]	0.44 [0.01, 1.25]	17
Muheza A, Tanzania 2006-07	2008 (CS)	13/39 [0.1-4.9]	30.14 [18.08, 42.53]	18
Muheza B, Tanzania 2006-07	2008 (CS)	164/671 [0.1-99.0]	23.82 [20.41, 27.37]	19
Muheza C, Tanzania 2006-07	2008 (CS)	417/1050 [0.4-19.9]	42.63 [39.44, 45.79]	19

**At each site, diverse sampling strategies were implemented. In Siaya A-C and Kilifi A-B, households underwent annual sampling as part of long-term surveillance. Kilifi C saw four rounds of household sampling, aligning with the hospital surveillance period. School surveys were conducted in various regions of Kenya (Busia, Kakamega, Vihiga, Bungoma, Homa Bay) and Uganda (Jinja A, Jinja B 2017-2018, Tororo A 2017-2019, Tororo B 2017-2019, Apac A & B, Mubende A & B, Kabale A & B). The survey included all public, primary schools within hospital catchment areas, with annual community-based household surveys also taking place. Community surveys in Jinja B and Tororo A and C (2012-2013) were part of broader household sample surveys across respective districts, with data limited to catchment parishes. Published findings in Muheza, Tanzania, provided village-level data reflecting time-site transmission estimates. Due to the varied sampling methods, the incorporation of sampling weights in PfPR₂₋₁₀ estimation was precluded. However, at all sites (excluding Muheza), samples covered entire community or school catchment areas within selected hospital catchment areas.*

***Parasite prevalence, adjusted for the 2-10 age range as detailed in methods is presented in bold. Estimates conducted by RDT were corrected to microscopy values using a regression framework outlined in Mappin et al. (2015)²⁰.*

A couple minor comments:

7. Figure 1C is nearly impossible to read

Thank you for this observation. We have now modified the figure aiming to improve its readability. Past prevalence estimates (i.e. median mPfPR) are plotted per each time-site either as yellow (for those sites where median prevalence has increased until the time of the hospitalization assessment) or as red (for those sites where median prevalence has reduced until the year of the hospitalization assessment). The arrows highlight the gap between past and present prevalence estimates and the direction of the change in the prevalence trend.

Fig 1 c

We have also modified the Figure caption as following:

“(c) Summarizing the prevalence trends over time estimated for each time-site as an increasing trend -when the estimated median $mPfPR_{2-10}$ in the past 7 to 9 years is lower than the $ePfPR_{2-10}$ at the time of assessment of severe outcomes incidence- or decreasing trend the estimated median $mPfPR_{2-10}$ in the past 7 to 9 years is higher than the $ePfPR_{2-10}$ at the time of assessment of severe outcomes incidence. Median past $mPfPR_{2-10}$ per each time-site is plotted as yellow (for those with increasing trends) or red (for those time-sites with reducing trends) and $ePfPR_{2-10}$ per time-site is plotted in orange. Estimates have been computed among the 27 time-sites with at least seven years of available past $mPfPR_{2-10}$ estimates. Black arrows indicate the direction of the change -increasing or decreasing- and the magnitude (as length).

8. The description of the $mPfPR$ is somewhat unclear. The authors note that this is a modeled estimate, as opposed to the empirical estimate ($ePfPR$). The authors then refer to this modeled estimate as median $PfPR$. Are they taking the median of the posterior from the model?

We define two sources of $PfPR$, the empirical and the modeled, which is an estimate computed for the specific catchment populations based on Allegana et al ⁹. That modeled estimates are computed per site and per year, with some sites with available estimates from 2000 to 2020 (e.g., Kilifi, Siaya, etc, see Supplementary Fig S1 for all sites) and others from 2010 to 2020, (e.g., Jinja, Tororo, etc). To estimate the “gap” between past cumulative transmission and present

transmission we then use the modeled estimates (as we do not have empirical information). To evaluate the gap, we compute for each site, the median PfPR across the annual PfPR estimates obtained for each year for the past 7 to 9 years . Here we included a minimum of 7 years estimates because not all time-sites have enough time point estimates, and a maximum of 9 because it comprises the maximum time for those children at their maximum age at the time of survey. We then compare this median estimate with the present-year empirical estimate to assess the relationship (whether the median of the modeled past transmission vs the empirical present transmission is significantly lower, higher or similar). The approach allows us to have a generalizable assumption on the change in the transmission patterns. Subsequently, we parameterize our model allowing simulations to show this type of transmission patterns.

We have now edit the Results section for clarification (line 97):

“For each time-site of the contemporary dataset , annual mPfPR₂₋₁₀ estimates were obtained using a Bayesian hierarchical geospatial model detailed elsewhere ^{8,9}. In those time sites where modeled estimates were available for at least 7 past years (n=27), and up to a maximum of 9 years, we computed the median mPfPR₂₋₁₀ of the time-series. We assume that the median prevalence across the past 7 to 9 years roughly represents the cumulative past transmission to which the population of children up to 10 years of age have been long-life, exposed which can be then compared to the empirical prevalence at the time of survey to evaluate the gap between past and present-day transmission.”

And again in line 137

“The difference between the ePfPR₂₋₁₀ and the median value of past mPfPR₂₋₁₀ can be interpreted as the gap in past exposure relative to ongoing exposure.”

Finally we now provide a more detailed description in the Methods section:

“We assessed the past exposure dynamics using catchment site-specific time-series of modeled age- and test-standardized parasite rates estimates, referred to as mPfPR₂₋₁₀ ^{8,9}. Modeled mPfPR₂₋₁₀ estimates were explicitly obtained for each time-site catchment population using a geospatial model detailed elsewhere ^{8,9}, a Bayesian hierarchical geostatistical framework based on more than 180000 geo-coded empirical prevalence survey data points from East Africa, interpolated in time to 1 x 1 km resolutions using climatic and ecological covariates. For a set of sites (n=11), annual mPfPR₂₋₁₀ estimates are available since 2000, while for the rest (n=15) available data begin in 2010. Time series of all the available mPfPR₂₋₁₀ per site are shown in fig. S1. Thus, for 27 out of the 35 present-day time sites, mPfPR₂₋₁₀ time series included at least seven time-points of past annual estimates. Further, to estimate the gap between past and present transmission, we first computed the median value of the annual mPfPR₂₋₁₀ time-series for each of the time-sites, under the assumption that it roughly represents the life-long transmission at which the surveyed population of children have been exposed in the previous years previous. This median mPfPR₂₋₁₀ can be then compared to the empirical PfPR₂₋₁₀ that is estimated at the time of surveying the malaria hospitalization incidence”

Reviewer #2 (Remarks to the Author):

This is a very interesting paper looking at the relationship between (time-varying) malaria exposure and severe disease risk. The authors re-examine an established contemporary dataset of malaria hospitalizations in children as well as historical data on hospitalizations, in conjunction with geospatial estimates of time-varying malaria prevalence and a mathematical model of malaria transmission. The authors show that discrepancies between the contemporary and historical datasets with respect to the shape of the severe incidence – prevalence relationship can be explained by recent changes in malaria exposure, whether that is a recent decrease or recent increase.

The authors have clearly thought deeply about the empirical observations, and I appreciate that the authors who originally published the contemporary dataset have continued to examine this data in detail to see what else can be learned from it. The result on steady vs unsteady transmission is, in retrospect, perhaps rather obvious but the authors have done a nice job providing strong support for it.

We thank the reviewer for the overall positive comment.

My main comment on this manuscript is that I found parts quite confusing and not clearly motivated, particularly in relation to the use of the mathematical model.

Minor comments:

9. L23-4: is there a difference between severe malaria and hospitalizations? It would be helpful if this work used one term exclusively, unless some aspects of the analysis were on severe malaria and some were on hospitalizations.

Thank you. This is an important point that we have discussed several times among the authors. Although our empirical variable (observation) is based on hospitalizations, the data is curated with inclusion/exclusion criteria that tries to reduce the rate of “false” severe malaria outcomes (i.e., by only considering cases where malaria is the primary cause of infection). We then assume that the curated data reasonably represents severe outcomes of malaria (see Paton et al for more specific details on the clinical definitions). For consistency we refer to hospitalizations when explaining or discussing our findings, but we now use the term of severe malaria outcomes when referring to broader concepts.

Thus, we have now modified the title as following:

“Severe outcomes of malaria in children under time-varying exposure”

In the Introduction, we include the following statement:

“We use community-based malaria hospitalization incidence rates in small catchment populations, and adjusted for case under-ascertainment, as an empirical proxy of the incidence of severe malaria outcomes.”

In the Results sections we have edited the following sentence for clarity:

“This allows us to compare and interpret our findings with a dataset that has been classically used for informing malaria inference models^{21–23} aiming to estimate severe outcomes of malaria across populations.”

We have edited the limitations in the Discussion section as follows:

“Our analyses have several limitations. First, we use community-based hospitalization in non-urban settings as an empirical proxy of the incidence of severe outcomes of malaria. While these estimates could under- or over-estimate the true number of severe outcomes, the data was obtained aiming to standardized the under-ascertainment of cases, estimates would be affected by site specific treatment-seeking behaviors and therefore represent a lower limit for hospitalization rate at each time site. Also, the empirical contemporary dataset does not necessarily represent urban settings, where the referral pathways from infection to hospitalization might be more complex to understand or subject to other potential bias. Further, the curated data does not include cases where malaria was not the major syndrome for hospitalization.”

10. L27: perhaps helpful to be clear that “unsteady” is in either direction (increasing or decreasing)

We agree with the reviewer. We have now clarify this concept in the Abstract and in the Results section line 163 :

“Here, we define steady exposure as the exposure of a specific population that does not change substantially over years, and we define unsteady exposure as an exposure that shows substantial increasing or decreasing dynamics over time.”

And in line 210:

“Given that contemporary model predictions based on steady exposure do not capture the pattern of the empirical contemporary data, we further evaluated scenarios with unsteady past malaria transmission (either increasing or decreasing dynamics), and how these different trends affect the PfPR₂₋₁₀-hospitalization rate incidence relationship”

And in the methods section:

“The relationship between the initial and final exposure rate was parameterized to emulate major patterns of the relationship as the longitudinal mPfPR₂₋₁₀ data, depicted in fig. 1b, and showing 1) substantial increase in the mPfPR₂₋₁₀ (e.g., Apac A), 2) relative constant mPfPR₂₋₁₀ (e.g, Busia), and 3) substantial decrease in the PfPR₂₋₁₀ (e.g., Jinja B). Thus, our definition of time-varying (unsteady) exposure comprises both substantial increasing and decreasing dynamics.”

11. L43-4: is the implication that inference accuracy depends only on understanding how clinical outcomes are related to exposure and not to other factors such as access to high-quality treatment?

With this statement, we wanted to highlight that models that aim to uncover the burden of malaria disease, whether they rely on simple or complex mathematical or computational sophisticated approaches always require data with sufficient quality as well as a good understanding of the key causal pathways that drive the relationship between the measured variable (in our case, prevalence) towards the estimated outcome (severe malaria outcomes). Unfortunately low quality data and in sufficient causal understanding, can not be solved with just using statistical and/or computational methods to produce generalizable interpretations. We have now clarified this matter in the Main section as following:

“Independent of the complexity of the approach, the ability of a model to generate accurate, robust and valid malaria disease outcomes using exposure predictors, such as prevalence, requires (1) high-quality data as input, from real-world observations, and (2) a comprehensive understanding and identification of the key factors determining the relationship between exposure and clinical outcomes.”

12. Fig 1a: should x axis label be ePfPR rather than mPfPR?

Thank you. We have corrected the error as follows:

13. Fig 1b: consider indicating the ePfPR on these plots

Thank you. We have now included the ePfPR estimates

And modified the figure caption consequently as follows:

(b) Present day $mPfPR_{2-10}$ over time in four representative time-sites (red, green, purple and blue colored lines) and time-site $ePfPR_{2-10}$ (orange) with highlighted time-periods for which hospitalization incidence estimates were available for the empirical relations in (a).

14. L115: the substantial decrease in Jinja B is based only on the 2010 $mPfPR$, is that correct? Why did some sites have $mPfPR$ going further back historically than others? Do we think the 2010 estimate reflects multiple years of high transmission before 2010, or could it be an outlier?

Thank you. We use the data that was available for each site. For some sites (Apac, Jinja, Mubende etc), available data begins in 2010 and for others (Busia, Kilifi etc) in 2000. Those are single point (i.e. single year) estimates, thus we assumed it reflects the average transmission at that year, including in Jinja B. For previous years there are no $mPfPR$ estimates for that site.

We have now clarified this in the Methods section:

“For a set of sites ($n=11$), annual $mPfPR_{2-10}$ estimates are available since 2000, while for the rest ($n=15$) available data begin in 2010. Time series of all the available $mPfPR_{2-10}$ per site are shown in fig. S1.”

15. L126-7: It would be helpful to motivate a little the approach of using the model to assess the impact of changing transmission. The current presentation seems to suggest the model is magic --- but I suspect there are good reasons to believe the model is capturing something (changing population immunity in response to changing exposure, and resulting changes in clinical presentation) accurately. Why would we think that? And therefore why would we consider using a model like that to better understand this severe dataset?

Thank you. We agree with the reviewer that the narrative is better built by contextualizing the approach. We have now included the following in the Results section:

“In addition to addressing the detailed major key changes between the historical and contemporary datasets, we further address the life-long exposure dynamics evidenced by the modeled community prevalence estimates (fig 1b and fig S1). The risk of malaria disease outcomes, including those severe, depend on immunity, which in turn depends on previous exposure dynamics. We hypothesize that the year-to-year variability on PfPR could strongly influence the risk of hospitalization. Thus, we further assessed the impact of this variability on malaria exposure on the ongoing hospital admission risk estimated from our individual-based model scenarios.”

16. L181: what are the representative patterns? This is clearer after reading the supplement, but I suggest also clarifying here whether you are simulating a generic step-up or step-down, or if the change is meant to be site-specific.

We have now clarified this in the Results section:

“Given that model predictions based on steady exposure do not capture the pattern of the empirical contemporary data, we further evaluated scenarios with unsteady past malaria transmission (either increasing or decreasing dynamics), and how these different trends affect the PfPR₂₋₁₀-hospitalization rate incidence relationship. Specifically, we performed simulations that included step-up and step-down exposure dynamics with differences between pre- and intra-survey prevalence within the range of those observed empirically (see Methods). Simulations captured representative patterns of time-varying exposure, with decreasing, steady or increasing transmission before computing severe disease incidence (fig. 2c) over the range of PfPR₂₋₁₀ values.”

In the Methods section, this is explained in detail:

“Particularly, we evaluated how a rapid reduction or increase of the PfPR₂₋₁₀ level over 6 months (e.g., PfPR₂₋₁₀ increasing from 10% to 70%) could affect the severe malaria incidence relative to a steady-state transmission assumption described before. This was performed by setting (1) the initial exposure rate (i.e., the individual probability of exposure to infectious bites pre-survey as a single rate over simulation time-steps), independent from (2) the final exposure rate (i.e., the probability of exposure during the survey period, also as a single rate over time steps),e. For simplicity, we set constant initial and final exposure rates. The relationship between the initial and final exposure rate was parameterized to reflect archetypical mPfPR₂₋₁₀ patterns, - depicted in fig.1b - showing (1) substantial increase in the mPfPR₂₋₁₀ (e.g., Apac A), (2) relative constant mPfPR₂₋₁₀ (e.g, Busia), and (3) substantial decrease in the PfPR₂₋₁₀ (e.g., Jinja B). Thus, our definition of time-varying (unsteady) exposure comprises both substantial increasing and decreasing dynamics. For each time-site with retrospective data encompassing at least 7 years, we computed the average mPfPR₂₋₁₀ available up to 9 years prior to the date when the hospitalization data was available, used it as a proxy of the initial exposure rate and computed

the fold-change PfPR₂₋₁₀. We then performed a local polynomial regression model to obtain predictions of the relationship between the final exposure rate and the corresponding relative change per time-site. See fig. S5 in supplementary depicting the relationship between the contemporary community prevalence as ePfPR₂₋₁₀, and estimated relative change at survey computed from the median value of mPfPR₂₋₁₀ of the past 7 to 9 years for each time-site. We then used these inputs of exposure to produce simulations using a combination of initial and final exposure rates to map a range of simulation-based PfPR₂₋₁₀ hospitalization rates.”

17. L272: sentence beginning “When the population immunity” I did not understand this at all

We agree this was confusing and not necessary as we had already explained it earlier in the paragraph. We have thus removed the section.

18. L349: “access rates” this refers to access to treatment? Perhaps re-word this sentence

We refer to access rates as the rate at which individuals either presenting uncomplicated malaria or severe malaria access to the syndrome-specific diagnostic and subsequent treatment. We have now clarified this as following in the Methods section:

“3) the rates of individuals suffering from malaria that required hospitalization who received diagnostic and treatment ranged from 60%-90% while the rate of individuals with uncomplicated malaria accessing diagnostic and treatment ranged 20%-60%²⁴”

19. Section beginning L347: were simulations / scenarios drawn from these options (parameters sampled from these ranges)? Or were they deliberately constructed? Based on the end of Section S4 I think they were deliberately constructed based on DHS and other data, is that right? Unclear from this section.

We agree that it is not sufficiently specified. The values used to parameterize the model are random samples obtained from a uniform probability of estimates that include the DHS estimates for the countries (Uganda, Kenya and Tanzania). We have now added for clarification the following in the methods section:

“For 2), 3) and 4), we parameterized the model by randomly sampling values from a uniform distribution defined by the respective ranges. Sensitivity analysis for these assumptions can be found in the Supplementary, Section S3.”

20. L393: out of curiosity...why is it necessary to simulate 90 years if you only look at children under 10? Why not simulate just 10 years?

We use 90 years to allow a “whole generation of individuals” to go through life-long malaria transmission (and let the ones infected at time 0 “die”) to ensure that the full age-structure of the

population that we evaluate reasonably represents reality. We have now clarified this in the Methods section as following:

“(i.e., over 90 years which ensures that lifelong malaria exposure has occurred all the generations evaluated prospectively)”

Supplement:

21. L193-8: *totally completely confused by this. Perhaps more specificity would be helpful? I'm not even sure what to ask that would help me understand.*

Thank you. In this paragraph we had used causal inference technical language (see for example, “Hernán MA, Robins JM (2020). *Causal Inference: What If*. Boca Raton: Chapman & Hall/CRC.”). We constructed a conceptual framework, based on causal reasoning, to ensure we approached systematically the assumptions that we would make to parameterize the model (i.e., we are not missing important relationships and evidence). However, we believe the very details of this framework are not fundamental to understand the work. We have now rewritten the paragraph in the supplement aiming to use less technical/causal language.

“The model structure of OpenMalaria is based on assumptions given prior scientific evidence and/or empirical observations on malaria transmission dynamics and pathogenesis. Before evaluating the contemporary and the historical empirical datasets, we used a conceptual framework based on causal inference²⁵ to systematically evaluate key assumptions we needed for designing the simulation scenarios, and to ensure consistency with the major changes that have occurred between the 90s (when the first dataset was collected) and a contemporary time in malaria clinical and epidemiological interventions. First, we defined the core causal pathways of the variables that defined the malaria outcomes, namely exposure, infection (for prevalence) and severe outcomes (for hospitalization incidence). We then defined and included the epidemiological factors -as exposures- that influence the process leading from exposure to infective mosquito bites towards infection and severe disease. We focus on factors that, if present, drive differences at the population level (such as exposure to a certain entomological inoculation rate) in opposition to those that lead to individual variability of severe disease risk (such as individual variability in becoming infected after a mosquito bite). We then formulated a simple regression model for the occurrence of severe disease at the population level as a function of malaria prevalence and included as predictors: 1) the overall immunity of the observed population of children, 2) the case management, which includes effective diagnostic and effective treatment, and 3) the occurrence of comorbidities influencing malaria severe risk. Thus, we assumed that the hospitalization risk among children is a function of the ongoing exposure, the life-long immunity elicited by the previous years exposure, the joint probability of receiving diagnostic and treatment and the efficacy of these diagnostic and treatment and the probability of co-occurring comorbidities. The conceptual framework allowed us to systematically address all the OpenMalaria parameters that relate to these statistical predictors.”

22. L200: *what does it mean to have a predictor “population-level immunity”?*

Here we refer to the level of immunity that a specific population (children in this case) has developed given their cumulative exposure to malaria. As the exposure might not be necessarily homogeneous across all the individuals (because of variability on age or spatial distribution of exposure), with the term we aimed to highlight that the immunity is that of the whole population (i.e. the average, median or mean with uncertainty range) rather than the individual immunity, which can be different between individuals. Given this and the previous comment, we have rewrite the sentence in the Supplementary as following:

“We then formulated a conceptual regression model for the occurrence of severe disease at the population level as a function of malaria prevalence and included as additional independent variables: 1) the overall immunity of the observed population of children, 2) the case management, which includes effective diagnostic and effective treatment, and 3) the occurrence of comorbidities influencing malaria severe risk. Thus, we assumed that the hospitalization risk among children is a function of the ongoing exposure, the life-long immunity elicited by the previous years exposure, the joint probability of receiving diagnostic and treatment and the efficacy of these diagnostic and treatment and the probability of co-occurring comorbidities. “

23. L251 section and Fig S2 and S3: this part also not clear to me. It's not easy to eyeball the figures and see what the authors are talking about here with age patterns of severe incidence by PfPR.

Thank you. The overall idea in this paragraph is that, although in the main findings we aggregate the population of children over age, when we look at the age disaggregation of the trends the model reasonably recovers the patterns, thus supporting the main findings. We can observe these patterns in two different ways, although they represent the same dynamics. In the first one (Fig S2), we use the same dimensions for plotting (hospitalization vs incidence across time-sites) but disaggregated by age-group. This allows us to see that the “exponential” pattern is present in the younger age groups as expected and does not change towards an asymptotic or convex pattern among the older ones as it would be expected under steady-transmission. Upon visual inspection, we observed that this empirical pattern is recovered by our model simulations, thus we conclude that the epidemiological processes leading to the age-dependent incidence are reasonably represented by our model and the assumption of unsteady transmission. When plotting the data the second way (Fig S3)- the standard way for plotting incidence over age- incidence is a function of age, and therefore we disaggregate by time-site (and thus also prevalence). Again, because the unsteady model-based age structure under the unsteady transmission recovers more accurately the empirical age structure, we consider that this supports our findings.

We have now rewritten the following paragraph in the Results for clarifications:

“Consistent with these results, our model recovers more accurately the observed hospitalization age structure (fig S3, left column for representative time-sites) under the

unsteady transmission assumption (fig S3, middle column) than under the steady-state assumption (fig. S3, right column)”

And clarified with more details in the supplementary as follows:

Age-structure of malaria severe disease under steady transmission is known to show characteristic dynamics, with low prevalence settings shifting the severe cases towards older children relative to higher transmission settings²⁶. As a result, severe malaria incidence among young and very young children in high transmission settings are typically higher than among same age-groups in low transmission settings whereas the opposite occurs among older children. When we examine the empirical patterns disaggregated by age-group (fig S2), we observed that the exponential relationship present in the younger ages is maintained over the older groups, although with less steepness. This is in opposition to what would be observed in clinical series with an asymptotic pattern (i.e., with steady transmission), as the relationship among older children would transition towards an asymptotic pattern (or even a convex one).

The simulations under the time-varying transmission assumption reasonably recover the trends across age-groups. Consistent with the main results here, we find that, for any age group, severe incidence increases towards higher PfPR2-10. Further, the empirical data shows that time sites with low PfPR2-10 present a lower-than-expected severe malaria risk for older ages. When analyzing these trends by looking at the incidence over age-groups disaggregated by time-site (fig. S3, plotting 4 representative time-sites) the empirical estimates of severe malaria occurrence over age-groups (left column) are better recovered under the time-varying assumption (center column) than under the steady- state transmission assumption. The overall analysis is consistent with an excess (e.g., fig. S3, Apac A) or reduction (e.g. fig. S3, Jinja B) of severe disease risk relative to the steady-state assumption, particularly towards older ages, as expected with time-varying exposure and subsequent gap between age-dependent developed immunity and transmission at assessment.

24. Fig S6: this figure was quite helpful to understand what the authors were doing. I would consider putting it in the main text if there is room, and perhaps also putting more detail on it (what assumptions are being used to parameterize the IBM? For example), and fitting it into the larger conceptual flowchart of the analysis approach.

Thank you. We have now changed the figure to include details on the assumptions that are used to parameterize the model. While we believe this is a useful figure to understand the modeling framework, we think it is still more consistent to keep it in the Supplementary. Nevertheless, for facilitating the understanding of the manuscript, we have now address this earlier in the Results sections:

To evaluate under which scenarios OpenMalaria can recover the empirical prevalence-hospitalization we implemented an iterative 4-step procedure, to explore hypotheses of impact of the determinants on disease risk and changes in disease risk – see sections above for details.

The procedure is represented schematically in the Supplementary (fig S6). For each scenario analysis, we performed four iterative steps as follows:

- Using a high-performance computing framework (<http://scicore.unibas.ch/>) we performed 1000 population-level individual-based model simulations of malaria transmission at a steady-state (i.e. same entomological inoculation rate for each simulation) over long periods of time (i.e., over 90 years which ensures that lifelong malaria exposure has occurred all the generations evaluated prospectively); computing hospitalization rates among children 3 months- 9 years per 1000 persons-year across values of PfPR₂₋₁₀ within the input range. In each of the simulations, we parameterized the model mapping the range of values set up for the major epidemiological determinants described in the previous section, namely disease management (which includes rates and efficacy of diagnostic and treatment of severe and uncomplicated malaria) and co-occurrence of comorbidities. We set the parameters according to the empirical dataset we were evaluating (i.e., historical or contemporary). At the same time, we evaluated the model outcomes simulating steady and unsteady transmission.

And the Supplementary figure 6:
Fig. S6.

Fig. S6. Schematic illustration of our iterative analysis approach to interrogating empirical data with mechanistic models

References

1. World malaria report 2023. <https://www.who.int/teams/global-malaria-programme/reports/world-malaria-report-2023>.
2. Weiss, D. J. *et al.* Mapping the global prevalence, incidence, and mortality of *Plasmodium falciparum*, 2000–17: a spatial and temporal modelling study. *Lancet* **394**, 322–331 (2019).
3. Guinovart, C. *et al.* The epidemiology of severe malaria at Manhica District Hospital, Mozambique: a retrospective analysis of 20 years of malaria admissions surveillance data. *The Lancet Global Health* **10**, e873–e881 (2022).
4. The DHS Program. <https://dhsprogram.com/methodology/survey-types/mis.cfm>.
5. Paton, R. S. *et al.* Malaria infection and severe disease risks in Africa. *Science* **373**, (2021).
6. Searle, K. M. *et al.* Household structure is independently associated with malaria risk in rural Sussundenga, Mozambique. *Front. Epidemiol.* **3**, 1137040 (2023).
7. Bannister-Tyrrell, M. *et al.* Importance of household-level risk factors in explaining micro-epidemiology of asymptomatic malaria infections in Ratanakiri Province, Cambodia. *Sci. Rep.* **8**, 1–15 (2018).
8. Kamau, A. *et al.* Malaria hospitalisation in East Africa: age, phenotype and transmission intensity. *BMC Med.* **20**, 28 (2022).
9. Alegana, V. A. *et al.* *Plasmodium falciparum* parasite prevalence in East Africa: Updating data for malaria stratification. *PLoS Global Public Health* vol. 1 e0000014 Preprint at <https://doi.org/10.1371/journal.pgph.0000014> (2021).
10. Muthui, M. K. *et al.* Gametocyte carriage in an era of changing malaria epidemiology: A 19-year analysis of a malaria longitudinal cohort. *Wellcome Open Research* **4**, (2019).
11. Wamae, K. *et al.* Transmission and Age Impact the Risk of Developing Febrile Malaria in Children with Asymptomatic *Plasmodium falciparum* Parasitemia. *J. Infect. Dis.* **219**, 936 (2019).
12. Kamau, A. *et al.* Malaria infection, disease and mortality among children and adults on the coast of Kenya. *Malar. J.* **19**, 1–12 (2020).

13. Were, V. *et al.* Trends in malaria prevalence and health related socioeconomic inequality in rural western Kenya: results from repeated household malaria cross-sectional surveys from 2006 to 2013. *BMJ Open* **9**, (2019).
14. Irimu, G. *et al.* Approaching quality improvement at scale: a learning health system approach in Kenya. *Arch. Dis. Child.* **103**, (2018).
15. Kanya, M. R. *et al.* Malaria transmission, infection, and disease at three sites with varied transmission intensity in Uganda: implications for malaria control. *Am. J. Trop. Med. Hyg.* **92**, (2015).
16. Staedke, S. G. *et al.* The PRIME trial protocol: evaluating the impact of an intervention implemented in public health centres on management of malaria and health outcomes of children using a cluster-randomised design in Tororo, Uganda. *Implement. Sci.* **8**, 1–13 (2013).
17. Mpimbaza, A. *et al.* The age-specific incidence of hospitalized paediatric malaria in Uganda. *BMC Infect. Dis.* **20**, (2020).
18. Ishengoma, D. S. *et al.* Trends of Plasmodium falciparum prevalence in two communities of Muheza district North-eastern Tanzania: correlation between parasite prevalence, malaria interventions and rainfall in the context of re-emergence of malaria after two decades of progressively declining transmission. *Malar. J.* **17**, 1–10 (2018).
19. Bernard, J. *et al.* Equity and coverage of insecticide-treated bed nets in an area of intense transmission of Plasmodium falciparum in Tanzania. *Malar. J.* **8**, (2009).
20. Mappin, B. *et al.* Standardizing Plasmodium falciparum infection prevalence measured via microscopy versus rapid diagnostic test. *Malar. J.* **14**, 460 (2015).
21. Ross, A., Maire, N., Molineaux, L. & Smith, T. An epidemiologic model of severe morbidity and mortality caused by Plasmodium falciparum. *Am. J. Trop. Med. Hyg.* **75**, 63–73 (2006).
22. Smith, T. *et al.* Mathematical modeling of the impact of malaria vaccines on the clinical epidemiology and natural history of Plasmodium falciparum malaria: Overview. *Am. J. Trop.*

Med. Hyg. **75**, 1–10 (2006).

23. Griffin, J. T. *et al.* Gradual acquisition of immunity to severe malaria with increasing exposure. *Proc. Biol. Sci.* **282**, 20142657 (2015).
24. The DHS Program. <https://dhsprogram.com/methodology/survey-types/mis.cfm>.
25. Hernan, M. A. & Robins, J. M. *Causal Inference*. (CRC Press, 2019).
26. Carneiro, I. *et al.* Age-patterns of malaria vary with severity, transmission intensity and seasonality in sub-Saharan Africa: a systematic review and pooled analysis. *PLoS One* **5**, e8988 (2010).

Reviewers' Comments:

Reviewer #1:

Remarks to the Author:

I thank the authors for adequately responding to this reviewer's comments. I have no further comments or suggestions.

Reviewer #2:

Remarks to the Author:

Thank you to the authors for a thorough job addressing these comments.

My one remaining comment is about my comment #15 on why use the model. This is still unclear in the revised text. Why would an individual-based model capture the relationship between exposure dynamics, immunity, malaria disease, and severe disease? It is not explicitly stated that the model in question includes XYZ features and has been calibrated / validated to XYZ empirical observations. What if I took a very basic SI (but individual-based!) model of malaria and tried to do what the authors have done? I would get meaningless outputs. This is what I meant, sorry if this has not been clear until now. Yes it is stated in the introduction that OpenMalaria has been previously validated, but there are specific ways in which it has been validated that make it suitable for this application.

Response to reviewers

REVIEWERS' COMMENTS

Reviewer #1 (Remarks to the Author):

I thank the authors for adequately responding to this reviewer's comments. I have no further comments or suggestions.

Thank you.

Reviewer #1 (Remarks on code availability):

The github link does not work

We have fixed the link. We believe now it works properly at:
<https://github.com/PDeSalazarSwissTPH/SevereMalaria>

Reviewer #2 (Remarks to the Author):

Thank you to the authors for a thorough job addressing these comments.

My one remaining comment is about my comment #15 on why use the model. This is still unclear in the revised text. Why would an individual-based model capture the relationship between exposure dynamics, immunity, malaria disease, and severe disease? It is not explicitly stated that the model in question includes XYZ features and has been calibrated / validated to XYZ empirical observations. What if I took a very basic SI (but individual-based!) model of malaria and tried to do what the authors have done? I would get meaningless outputs. This is what I meant, sorry if this has not been clear until now. Yes it is stated in the introduction that OpenMalaria has been previously validated, but there are specific ways in which it has been validated that make it suitable for this application.

Thank you. Within the modeling tools in epidemiology, individual-based models allow to increase the weighting of theory relative to the available data by combining empirical observations with essentially unverifiable data (theory) as a scientific collage. This allows to evaluate more upstream interventions for which randomized experiments are not possible or can not be emulated, and to evaluate interventions in complex systems for which data to produce informative estimates do not exist ¹. On the other hand these models can not be tested experimentally, and thus generally accepted principles are set to build the models, populate and calibrate their parameters, and test their predictions to avoid model misspecification.

Individual-based models, if approximately correct, can therefore describe systems that exhibit dynamically complex properties, such as multiple interacting causal effects, feedback loops, threshold dynamics and interference ². These are the characteristics that made individual-based

modeling the most suitable venue for evaluating the complex dynamics between exposure, immunity, and clinical outcomes at a metapopulation level in malaria epidemiology, and the complex system of effects that influence those variables in complicated manners. We have now clarify this in the results section as suggested by the reviewer:

“Understanding the complex relationship between malaria exposure, immunity and clinical outcomes across populations and time requires causal analytical frameworks which (a) can combine empirical observations with theory (b) can address multiple interacting causal effects, interference, threshold dynamics and interference (c) have generally accepted principles to build the models, populate and calibrate their parameters and test their predictions for avoiding misspecification. Individual-based models are amongst the few modeling tools that fulfill these requirements^{1,2} .

Open-Malaria (<https://github.com/SwissTPH/openmalaria/wiki>) is an multi-level individual-based model that includes several key features that allows to generate counterfactuals of the effect of malaria exposure on clinical disease under different scenarios of population structure, changing transmission, health-access, diagnostic thresholds, drug-efficacy, and other major malaria control and prevention interventions³. Random effects can be incorporated in the modeled processes and allow incorporating uncertainty and heterogeneity in the simulations. Relevant to our analyses, the framework encompasses submodels specifically parameterized to empirical data including (1) population structure^{4,5} (1) within-host dynamics of parasite burden and addressing the effect on single individuals of repeated infections in developing immunity and subsequent infections⁶⁻⁸; (2) disease progression^{6,9} and health-seeking behavior including rates of individuals accessing health services, as well as time to diagnostic and treatment¹⁰; (3) efficacy of case-management including diagnostic sensitivity and specificity, first- and second-line treatment effectiveness and efficacy of hospitalization¹⁰; and (4) the effect of age-structured comorbidities on severe malaria outcomes upon infection⁵ . Further details are provided in the supplementary Section S2”

Reviewer #2 (Remarks on code availability):

Please find the code at <https://github.com/PDeSalazarSwissTPH/SevereMalaria>

References

1. Hernán, M. A. Invited Commentary: Agent-Based Models for Causal Inference—
Reweight Data and Theory in Epidemiology. *Am. J. Epidemiol.* **181**, 103 (2015).
2. Marshall, B. D. L. & Galea, S. Editor’s choice: Formalizing the Role of Agent-Based
Modeling in Causal Inference and Epidemiology. *Am. J. Epidemiol.* **181**, 92 (2015).
3. Reiker, T. *et al.* Emulator-based Bayesian optimization for efficient multi-objective

- calibration of an individual-based model of malaria. *Nat. Commun.* **12**, 7212 (2021).
4. Ekström, A. M. *et al.* INDEPTH Network: contributing to the data revolution. *The lancet. Diabetes & endocrinology* **4**, (2016).
 5. Ross, A., Maire, N., Molineaux, L. & Smith, T. An epidemiologic model of severe morbidity and mortality caused by *Plasmodium falciparum*. *Am. J. Trop. Med. Hyg.* **75**, 63–73 (2006).
 6. Smith, T. *et al.* An epidemiologic model of the incidence of acute illness in *Plasmodium falciparum* malaria. *Am. J. Trop. Med. Hyg.* **75**, 56–62 (2006).
 7. Collins, W. E. & Jeffery, G. M. A retrospective examination of the patterns of recrudescence in patients infected with *Plasmodium falciparum*. *Am. J. Trop. Med. Hyg.* **61**, 44–48 (1999).
 8. Golumbeanu, M. *et al.* Leveraging mathematical models of disease dynamics and machine learning to improve development of novel malaria interventions. *Infect Dis Poverty* **11**, 61 (2022).
 9. Ross, A. & Smith, T. The effect of malaria transmission intensity on neonatal mortality in endemic areas. *Am. J. Trop. Med. Hyg.* **75**, (2006).
 10. Tediosi, F. *et al.* An approach to model the costs and effects of case management of *Plasmodium falciparum* malaria in sub-saharan Africa. *Am. J. Trop. Med. Hyg.* **75**, 90–103 (2006).